# Neurosymbolic Object-Centric Learning with Distant Supervision

## Abstract

Relational learning enables models to generalize across structured domains by reasoning over objects and their interactions. While recent advances in neurosymbolic reasoning and object-centric learning bring us closer to this goal, existing systems rely either on object-level supervision or on a predefined decomposition of the input into objects. In this work, we propose a neurosymbolic formulation for learning object-centric representations directly from raw unstructured perceptual data and using only distant supervision. We instantiate this approach in DeepObjectLog, a neurosymbolic model that integrates a perceptual module, which extracts relevant object representations, with a symbolic reasoning layer based on probabilistic logic programming. By enabling sound probabilistic logical inference, the symbolic component introduces a novel learning signal that further guides the detection of meaningful objects in the input. We evaluate our model across a diverse range of generalization settings, including unseen object compositions, unseen tasks, and unseen number of objects. Experimental results show that our method outperforms neural and neurosymbolic baselines across the tested settings.

## 1 Introduction

Relational learning has long served as a cornerstone of machine learning approaches that aim to generalize across structured domains. From Statistical Relational Learning (SRL) to Graph Neural Networks (GNNs), these paradigms offer a powerful mechanism for modeling interactions among entities—without being tied to fixed input dimensionality. By operating over graphs, sets, or logic-based representations, relational models achieve size-invariance and strong inductive generalization, leveraging local patterns such as message-passing rules or logical constraints that apply uniformly across varying input structures. For instance, a GNN trained on small molecule graphs can generalize to larger, unseen molecular structures by reusing local chemical interaction patterns. Such systematic generalization is a prerequisite for building AI systems that can adapt beyond their training distributions and reason flexibly in open-ended environments.

Neurosymbolic AI, which combines neural networks with logic reasoning, has been recently analysed (Marra et al., 2024) in terms of SRL dimensions, showing it inherits the relational strengths of its predecessors while incorporating a deep representation learning component. However, just like SRL, neurosymbolic frameworks (Manhaeve et al., 2018; Skryagin et al., 2022) make a critical assumption: *that the input is already structured*. The entities and their relationships are assumed to be given, often represented as a knowledge graph or a logical database, potentially extended with object-level subsymbolic representations (e.g. images). This assumption becomes a significant limitation when dealing with raw perceptual data, where such a structure must be inferred rather than assumed. In many real-world settings, inputs do not come with explicit object boundaries or relational structure; instead, this information must be inferred from complex, unstructured data, often under supervision that is incomplete or only indirectly related to the underlying relational structure. Neurosymbolic models must therefore bridge the gap between perception and reasoning by constructing symbolic representations directly from raw input. This ability to induce structure from subsymbolic data is arguably the core promise and the main challenge of neurosymbolic AI.

Object-centric learning (De Sousa Ribeiro et al., 2024) offers a promising foundation for structured representation learning. Models such as MONet, IODINE, and Slot Attention decompose visual scenes into object-like components, enabling representations that support forms of systematic gener-

Figure 1: Overview of the proposed model, DeepObjectLog, which integrates object-centric perception with symbolic reasoning in an end-to-end trainable architecture.

alization (Burgess et al., 2019; Greff et al., 2019; Locatello et al., 2020). While core object-centric models can be trained without annotations, pipelines that use these models for downstream semantic tasks often introduce object-level supervision to map each discovered slot to a symbolic category. This restricts their applicability when supervision is unavailable or impractical.

In contrast, neurosymbolic approaches seek to operate in *weakly supervised regimes*, where only global labels or logical constraints are available as training signals (Manhaeve et al., 2018). In these settings, models must not only parse unstructured inputs into symbolic entities but also do so in a way that supports downstream reasoning, even when the structure is ambiguous and no direct supervision over object identity or count is provided. For instance, a model trained to predict the sum of digits in an image without access to the digit labels or positions must learn to decompose the input in a way that supports correct logical inference. A total sum of 5 could correspond to many different groupings: a single digit "5", a pair like "2+3", or even five "1"s. The ambiguity in such supervision provides little direct signal about the number of entities present, yet the model must still learn a decomposition useful for reasoning. This paper argues that neurosymbolic learning must go beyond simply layering symbolic reasoning on top of neural models. It must also address the precondition for relational modeling itself: the detection of entities and their relations from unstructured input, under minimal supervision. This requires an integration of object-centric learning and relational reasoning not as sequential modules, but as jointly-trained components that inform each other. Below we highlight the problem we are trying to solve:

---

**Problem. [*Generalizable object-centric neurosymbolic learning from distant supervision*]**

*Setup*: We are given images that contain multiple objects, but the images come with no information about what the objects are, where they are, or how many there are. Instead of detailed object labels, we only have high-level supervision in the form of a global label and a set of logic rules describing how the unknown objects might relate to each other and contribute to the global label.
*Goal*: Learn a neurosymbolic model that can discover meaningful object-level structure from raw images in relational reasoning tasks, which require generalization beyond the training distribution, handling new combinations of objects, different numbers of objects, and entirely new reasoning tasks (i.e. new rules and classes).

---

In this work, we make several contributions. First, we highlight the limitations of current neural and neurosymbolic systems in achieving fully relational learning from unstructured input. Second, we propose a new probabilistic formulation for learning object-centric representations directly from raw perceptual data, enabling a model to identify, classify, and reason about multiple objects using only distant supervision from discriminative tasks. Third, we instantiate this formulation in DeepObjectLog, a model that integrates object-centric neural perception with symbolic reasoning based on ProbLog, allowing end-to-end object-centric learning driven by logical structure. Finally, we introduce benchmarks based on MultiMNIST, CLEVR, and the newly developed PokerRules dataset,

designed to assess the ability of models to discover relevant objects and reason over their relations. Experimental results show that our approach consistently outperforms both neural and neurosymbolic baselines across a range of generalization settings. A graphical overview of the proposed model is provided in Figure 1, while a detailed explanation of its components is given in Section 3.

## 2 RELATED WORK

The problem addressed in this study lies at the intersection of two research areas: object-centric representation learning and neurosymbolic reasoning. Both are essential, but neither is sufficient on its own to meet the demands of a system capable of learning relational abstractions directly from raw input when supervision is provided only at the task level.

**Object-centric learning.** Object-centric models aim to decompose visual scenes into discrete object representations (Greff et al., 2020). Early approaches (Burgess et al., 2019; Greff et al., 2019; Engelcke et al., 2019) use iterative variational inference to discover object structure in an unsupervised manner, but often suffer from scalability limitations. Slot Attention (Locatello et al., 2020) replaces these procedures with a parallel attention-based binding mechanism, enabling permutation-invariant inference over latent slots. This model has since been extended across several domains (Kori et al., 2024a; Elsayed et al., 2022; Seitzer et al., 2022; Singh et al., 2022), with slot-based representations being applied to generative modeling and dynamic scenes. Some works attempt to improve the selection of useful slots by adaptively selecting which representations carry meaningful content (Fan et al., 2024; Engelcke et al., 2021), but these still rely on heuristics or sparsity priors that are disconnected from task-level semantics. A recent study (Kori et al., 2024a) shows that slot-based models can be trained end-to-end with supervision on a downstream task, but it assumes a fixed number of objects and employs a neural classifier without symbolic reasoning, requiring few-shot learning to achieve task generalization. Without an additional task-level or logical supervision signal, current object-centric models learn object representations independently of the symbolic logic structure of the task, which limits their ability to identify objects, their relations, and their task-specific contributions in a way that aligns with human interpretation. These limitations are reflected in their constrained out-of-distribution generalization capabilities (Wiedemer et al., 2023).

**Neurosymbolic systems.** A complementary limitation appears in most neurosymbolic (NeSy) systems: while they reason over structured entities, they assume that this structure is already available. Thus, they cannot learn to infer object-centric representations from raw data. Neurosymbolic approaches integrate neural perception with symbolic logic to support probabilistic reasoning over structured representations (Marra et al., 2024; Manhaeve et al., 2018). These frameworks offer formal semantics and the ability to encode prior knowledge through rules, with neural modules supplying probabilistic predictions to the logic. However, in most cases, the input is assumed to consist of pre-segmented objects (e.g. individual images or attribute vectors) bypassing the challenge of discovering structure from raw perceptual input (Yi et al., 2018; Mao et al., 2019). Extensions to these frameworks incorporate limited spatial reasoning or component-level architectures that attempt to allow compositional and task generalization, but they typically operate with a fixed number of entities (Misino et al., 2022; De Smet et al., 2023). As a result, current neurosymbolic systems are not designed for object-agnostic parameterization. In Stammer et al. (2021); Shindo et al. (2023), the authors employ a pretrained Slot Attention module to obtain object-centric representations, subsequently using these for set prediction and rule extraction. In Skryagin et al. (2022), the authors further explore integrating logical constraints into the training of object-centric modules, demonstrating that logical supervision can improve object decomposition by introducing assumptions of independence across attributes. Finally, some works in abductive learning have attempted to learn object detection from unstructured data, but they typically rely on knowing in advance how many objects are present in each image, which limits their ability to fully handle unstructured inputs (Gao et al., 2024; Cai et al., 2021). While these approaches demonstrate the potential of combining perception with logic, they still rely on object-level supervision, such as object labels or question-answer pairs, to align neural representations with symbolic targets.

## 3 METHOD

To address the limitations outlined above, we introduce DeepObjectLog, a neurosymbolic architecture designed to reason on objects, their properties and their compositions, directly from an unstructured input and learning only from weak supervision. The model unifies object-centric perception, probabilistic inference and symbolic reasoning within a single, end-to-end trainable system. Its design explicitly satisfies the core requirements for relational neurosymbolic learning: it handles a variable number of objects, supports reasoning under uncertainty, and enables the emergence of relational structure without the need of an object-level supervision.

From a high level, the architecture is composed of the following components: (i) **Objects encoder**: starting from a global representation $z$, the model extracts up to $N$ latent representations $s_i$, each intended to describe a potential object in the scene; (ii) **Object detector and classifier**: for each extracted object encoding, the model infers a probability $\beta_i$ that the representation corresponds to a meaningful object, from which an *objectness* flag $o_i$ and a class $c_i$ are extracted; (iii) **Probabilistic Logic Reasoning**: the system reasons logically over detected objects and their classes to derive a downstream task. This enables the system to interpret objects, understand their relationships, and constrain object configurations in ways that support task-specific inference. (iv) **Image decoder**: finally, each representation is decoded into image space.

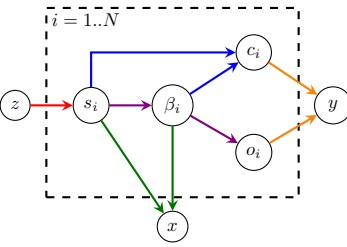

Figure 2: Probabilistic graphical model of DeepObjectLog

### 3.1 PROBABILISTIC GRAPHICAL MODEL

Let $z \in \mathbb{R}^{d_z}$ be a latent representation of the image. For each potentially present object $i$, we denote its associated slot as $s_i \in \mathbb{R}^{d_s}$, and the probability that it represents an actual object as $\beta_i \in [0, 1]$. The corresponding objectness flag is $o_i \in \{0, 1\}$ and the class label corresponding to the object is $c_i \in C \subset \mathbb{N}$. Finally, let $y \in Y \subset \mathbb{N}$ be the downstream class label computed by a symbolic logic theory $L$ and $x \in [0, 1]^{c \times h \times w}$ be the normalized image. The generative model in Figure 2 factorizes as:

$$p\big(y, x, \mathbf{s}, \boldsymbol{\beta}, \mathbf{o}, \mathbf{c}, z\big) \ = \ p(z)\, p_L\big(y \mid \mathbf{o}, \mathbf{c}\big)\, p\big(x \mid \mathbf{s}, \boldsymbol{\beta}\big) \Big[ \prod_{i=1}^{N} p(s_i \mid z) p(\beta_i | s_i)\, p(o_i \mid \beta_i)\, p(c_i \mid \beta_i, s_i) \Big]$$

(1)

where $\mathbf{s} = (s_1, \ldots, s_N)$, $\boldsymbol{\beta} = (\beta_1, \ldots, \beta_N)$, $\mathbf{o} = (o_1, \ldots, o_N)$, $\mathbf{c} = (c_1, \ldots, c_N)$.

This joint distribution consists of the following components:

- $p(z)$ a prior distribution of the latent representation of the scene;
- $p(s_i \mid z)$: the *slot extractor*, which produces the $i$-th latent slot representation $s_i$ out of $N$ maximum possible objects based on the latent representation $z$. A slot $s_i$ represents a *proposal* for a potential object.
- $p(\beta_i \mid s_i)$ models the uncertainty of $s_i$ being a meaningful object (and not, e.g., part of the background); it is represented explicitly since object-centric models can condition their processing on such uncertainty (Fan et al., 2024);
- $p(o_i \mid \beta_i)$ models the binary flag corresponding to the uncertainty $\beta_i$,
- $p(c_i \mid \beta_i, s_i)$: the symbolic classification head, which assigns a discrete class label $c_i$ to the representation $s_i$, conditioned on its uncertain objectness $\beta_i$;
- $p_L(y \mid \mathbf{o}, \mathbf{c})$: the symbolic reasoning component, which predicts the final task-specific output $y$ by reasoning on a logic theory $L$, the inferred objects $o_i$ and their classes $c_i$;
- $p(x \mid \mathbf{s}, \boldsymbol{\beta})$: the image generator, a probability distribution over images given the set of slots and their predicted objectness.

This probabilistic formulation allows the model to operate under the inductive assumptions that (1) object-level structure is latent and uncertain, (2) symbolic abstraction ($o_i$, $c_i$, $y$) and perceptual

grounding ($x$) are complementary flows, (3) relational reasoning can be performed over learned representations and (4) direct supervision at the object level can be avoided as it can be inferred from distant observations on $x$ and $y$. As we will show in the following sections, while the model depends on the hyperparameter $N$ for the maximum number of objects, each module is independent of $N$ and the actual number of objects, which is dynamically extracted by the objectness variables. Each slot will share the same modules, allowing the model to adapt to different number of objects both at training and inference time, and allowing to test the model in scenes with a number of objects that is different than the one used during training. This is a fundamental yet only partially fulfilled property in current neurosymbolic models.

Each of the components in the joint distribution corresponds to a functional module in the architecture. In the sections below, we describe the main components of the system in more detail, clarifying their role and interaction.

### 3.2 OBJECT INDEPENDENT ENCODER IN THE DISCRIMINATIVE SETTING

In this paper, we are interested in a discriminative setting, where we focus on modeling the probability distribution of the object classes $c_i$ by observing the task label $y$ (see Section 4), and using the image $x$ to amortize inference. Therefore, similarly to Kori et al. (2024b), we model the slot encoder distributions $p(s_i|\cdot)$ as deterministic delta distributions. Such an encoder should avoid relying on supervision or assumptions about object count or location, ensuring flexibility across inputs and generalization to novel object numbers and positions. Many possible existing models satisfy such requirements (Burgess et al., 2019; Engelcke et al., 2019) but we leave their comparison for future work. In our implementation, this module is instantiated using state-of-the-art Slot Attention (Locatello et al., 2020; Zhang et al., 2023), a differentiable mechanism for object-centric representation learning. In particular, $p(s_i|z) = SlotAttention_i(z; \theta_s, N)$ where $\theta_s$ is the real set of parameters of the slot attention model. Notice that the parameterization of the slot attention is independent of the maximum number of objects, which is regarded as a hyperparameter. Slot Attention produces a set of latent representations that aim at capturing distinct parts of the scene, and are well suited for further object-level processing. Therefore, the encoder provides a proposal decomposition of the input into object-like components for the subsequent classification stages.

### 3.3 OBJECT DETECTOR AND CLASSIFIER

The object detection is modelled over two variables: the uncertainty scores $\beta_i \in [0, 1]$ and the objectness flags $o_i \in \{0, 1\}$. The meaning is that $o_i$ is 1 (i.e. it represents an object) with probability $\beta_i$. We model directly the uncertainty $\beta_i$ over $o_i$ as many components in current state-of-the-art object-centric models exploit soft masking schemes using such uncertainty. However, we do not model any probability distribution over $\beta_i$; therefore $p(\beta_i|s_i) = \delta(\beta_i - \beta_i^*)$ is a delta distribution centered around the output value of a neural network $\beta_i^* = f_\beta(s_i|\theta_\beta)$ with $\theta_\beta$ its set of weights. We model $p(o_i|\beta_i)$ as a Bernoulli distribution, parameterized by $\beta_i$, i.e. $p(o_i|\beta_i) = \beta_i$ . The classifier $p(c_i|s_i, \beta_i)$ can be parameterized with either a Bernoulli or with a Categorical distribution, depending on the particular object classification task. In both cases, the distribution is parameterized by a neural network $f_c(\beta_i s_i; \theta_c)$ with weights $\theta_c$.

Notice that, following a similar approach to the one proposed in Fan et al. (2024), the slot representation $s_i$ in the input to the network $f_c$ is also multiplied elementwise by the uncertainty $\beta_i$. This gating mechanism allows the model to downweigh uncertain or irrelevant representations. Together, $p(o_i|\beta_i)$ and $p(c_i|s_i, \beta_i)$ allow the system to identify and categorize a variable number of object candidates, while explicitly modeling uncertainty about their presence and identity.

### 3.4 PROBABILISTIC LOGIC REASONING OVER THE TASKS

The final symbolic prediction $y$ is computed using probabilistic logical reasoning over the set of inferred object class predictions and objectness probabilities. This component is implemented using ProbLog (De Raedt et al., 2007), a probabilistic logic programming framework that extends classical logic programming with uncertainty modeling through probabilistic facts. While ProbLog is Turing-equivalent and allows for highly expressive probabilistic modeling, in this work we focus only on the

aspects relevant to neurosymbolic object-centric learning and limit our explanation accordingly. For a full overview of the framework, we refer interested readers to De Raedt et al. (2007).

**Probabilistic Logic Programming.** A ProbLog program $L$ is defined over two sets of syntactic constructs: probabilistic facts and rules. A probabilistic fact with syntax `p::f` is an independent Bernoulli distribution $p(f)$ over a binary variable `f` with parameter $p = p(f)$. For example, `0.1::alarm` means that variable `alarm` has 0.1 probability of being True. A rule is of the form `h :- b₁,...,bₘ`, where `h` is the head or conclusion and each `bᵢ` is a body element or premise. Rules act as definition rules: if all the premises $b_i$ are True, then the conclusion `h` must be True. For example, in the classical Pearl's burglary example, the rule "`call :- alarm, hear`" encodes that if the `alarm` goes off and we `hear` it, then we `call` the police. Given the set of all probabilistic facts $\mathbf{f}$ and the set of all possible rules $R$, ProbLog allows to efficiently compute the marginal probability of every symbol $y$ in the program as:

$$p(y) = \sum_{\mathbf{f}} p(y|\mathbf{f}; R) \prod_{f_i \in \mathbf{f}} p(f_i) \tag{2}$$

where $p(y|\mathbf{f}; R)$ is a deterministic distribution stating whether $y$ can be obtained from $\mathbf{f}$ when applying the rules in $R$ (possibly chaining multiple of them).

**ProbLog for object-centric learning.** We use ProbLog to model three components of our model:

- The object detectors $p(o_i|\beta_i)$ are modelled as probabilistic facts `object(i)` stating whether the i-th extracted slot is an object;
- The object classifiers $p(c_i|s_i, \beta_i)$ are modelled as probabilistic facts `class(i)` (for a binary classifier) or `class(i,k)` for a categorical classifier with class $k$. The conditional nature on the objectness $o_i$ is encoded using rules. The rule `task:-object(1),class(1,3)`, for example, states that `task` is true only if the 1st slot is an object and its class is 3.
- The conditional task distribution $p_L(y|\mathbf{c}, \mathbf{o})$ is the deterministic distribution obtained by all the logic rules proving the task $y$ given the objectnesses and the classes of all objects.

**Example 1 (Addition of at most two digits)** *Let us consider a setting where images contain zero to two digits that can only be $0$ or $1$. Consider the following two kinds of probabilistic facts. First, $\mathbf{p}_{o_{id}} :: $ `object(ID)` stating that the probability that the ID-th slot of the input image contains an actual digit is $\mathbf{p}_{o_{id}}$, as computed by $p(o_i|\beta_i)$. Second, $\mathbf{p}_{c_{id}} :: $ `class(ID,C)` providing the probability that the ID-th object is of class $C$, as computed by $p(c_i|s_i, \beta_i)$. We can encode the digit addition task in the following program:*

```
digit(ID, Val) :- object(ID), class(ID, Val).
digit(ID, 0) :- \+ object(ID).
add(Z) :- digit(1, Y1), digit(2, Y2), Z is Y1 + Y2.
```

*We can then query, for example, what is the probability that a given image with multiple digits sums to 2, i.e. $y = $ `add(2)`, given the distributions of the current objectness of the slots and their corresponding classes.*

Note that any marginal task distribution $p(y|\mathbf{s}, \boldsymbol{\beta})$, can be computed by standard marginal inference (Eq. 2) of the task query $y$:

$$p(y|\mathbf{s}, \boldsymbol{\beta}) = \sum_{\mathbf{c}, \mathbf{o}} p(y|\mathbf{c}, \mathbf{o}) \prod_i p(o_i|\beta_i) p(c_i|s_i, \beta_i) \tag{3}$$

It is interesting to note that the use of probabilistic logic eliminates the need for set matching losses typically employed in object-centric systems (e.g., Hungarian loss), as object assignments are implicitly resolved through marginalization.

## 3.5 MASKED DECODER

For the design of the image decoder $p(x|\mathbf{s}, \boldsymbol{\beta})$, we follow recent literature in Slot-attention architectures. In particular, we model the image as a mixture of Multivariate Gaussian distributions.

Each component is a multivariate Gaussian distribution $\mathcal{N}(\mu, I)$ with unit covariance matrix $I$ and a mean vector $\mu$ parameterized with a neural network $f_x(s_i\beta_i; \theta_x)$ with $\theta_x$ its parameters. The mixing weights $w$ of the components are also parameterized by a neural network $f_w(s_i\beta_i; \theta_w)$. Notice, as we did for the objectness variable $o_i$ and as commonly done in the slot-attention literature, we decode each slot $s_i$ after a soft masking using $\beta_i$.

# 4 LEARNING

Learning in DeepObjectLog aims at maximizing the marginal likelihood of pairs constituted by an input image $x$ and a task label $y$, without any direct supervision on the objects contained in the image and their classes. Let $D = \{x, y\}_K$ be a dataset of $K$ input-output pairs $(x, y)$, learning is the solution of the following optimization problem:

$$\max_{\theta_p} \sum_{x,y \in D} \log p(x, y; \theta_p) \tag{4}$$

where $\theta_p = \{\theta_s, \theta_\beta, \theta_c, \theta_x, \theta_w\}$ is the set of all the parameters of the joint $p(y, x, \mathbf{s}, \boldsymbol{\beta}, \mathbf{o}, \mathbf{c}, z)$, while $p(x, y; \theta_p)$ is the marginal $p(y, x) = \sum_{\mathbf{o}, \mathbf{c}} \int dz d\mathbf{s} d\boldsymbol{\beta}\ p(y, x, \mathbf{s}, \boldsymbol{\beta}, \mathbf{o}, \mathbf{c}, z)$. By exploiting the sifting property of delta distributions and the task marginalization of ProbLog (Equation 3), we have:

$$p(y, x) = \int dz\ p(x, y, z) = \int dz\ p(z)p(x|\mathbf{s}^*(z), \boldsymbol{\beta}^*(z))p(y|\mathbf{s}^*(z))$$

where $\mathbf{s}^*(z)$ and $\boldsymbol{\beta}^*(z)$ are the centers of $p(s_i|z)$ and $p(\beta_i|s_i)$, respectively, and both dependent on $z$.

Since we are interested in a discriminative setting only, we approximate the inference on the intractable marginalization by amortized MAP inference. In particular, we approximate the MAP state $z^* = \arg\max_z \log p(x, y, z)$ by a neural network encoder $z^* \simeq q(x; \phi)$ parameterized by $\phi$. Under such approximation, we compute a lower-bound of the max log-likelihood of Equation 4 as (Appendix A):

$$\max_{\theta_p, \phi} \sum_{(x,y) \in D} \log p\big(y|\mathbf{s}^*(q(x)), \boldsymbol{\beta}^*(q(x))\big) + \log p\big(x|\mathbf{s}^*(q(x)), \boldsymbol{\beta}^*(q(x))\big) + \log p\big(q(x)\big) \tag{5}$$

where the first addend is a maximum likelihood term for the observed label $y$, the second addend is a reconstruction term of the observed $x$ and the last addend is an activation regularization of $q(x)$.

# 5 EXPERIMENTS

In this study, we focus on evaluating the generalization capabilities of DeepObjectLog, as generalization under distributional shifts offers the clearest evidence that a model is truly learning to decompose scenes into meaningful components and reason over them (Greff et al., 2020; Dittadi et al., 2021). To guide our analysis, we pose the following research questions:

**Q1: Compositional generalization:** Can the model recombine familiar objects in novel ways beyond those encountered during training?
**Q2: Task generalization:** Can the model generalize to new tasks not present in the training data?
**Q3: Object count generalization:** Can the model detect, classify, and reason over object configurations larger (extrapolation) or smaller (interpolation) than those seen during training?
**Q4: Foundation models**: How do pretrained Vision-Language Models (VLMs) compare to our approach on these structured reasoning tasks?

## 5.1 SETUP

**MultiMNIST-Addition Dataset** (MM-A). We build a dataset containing 128×128 images, each containing a number of non-overlapping random MNIST digits with equal probability. The label associated with each image is the sum of the digits it contains, and no additional supervision is provided. To test compositional generalization, we build a dataset where the training and validation sets consist of images containing 0 to 3 digits and cover 75% of all possible digit combinations, while the test set includes the remaining 25%. To test extrapolation, we create two additional test sets where

a structural shift is introduced: each image contains 4 or 5 digits (more digits). In the first, to allow a comparison with all baselines, we constrain the digit combinations such that the resulting sums are the same ones as seen during training (no new classes). In the second, we additionally introduce a semantic shift by allowing higher digit sums (new classes). Finally, we include a last test set for measuring interpolation, training models on inputs with 0, 1, or 3 digits and testing them on images with exactly 2 digits.

**More complex scenarios.** We evaluate our approach on two additional datasets designed to test reasoning in more challenging settings and images. The **PokerRules dataset** consists of 256×256 images of horizontally arranged playing cards on a fixed background. Each image contains between 1 and 4 cards during training, with slight random spatial shifts to introduce variability. Labels correspond to the poker hand formed by the visible cards (e.g., pair). Generalization is assessed both on unseen hand types (new classes) and across extrapolation settings involving new numbers of cards. The **CLEVR-Addition Dataset** (CLEVR-A) is derived from CLEVR (Johnson et al., 2017), where we select images containing at most six objects. We then construct pairs of images and assign as label the sum of the number of objects present across the two images. Extrapolation is tested by including images with higher object counts than those seen during training, resulting in both in-distribution and out-of-distribution classes. For additional details, we refer to Appendix B.

**Baselines.** We compare our approach against four neural baselines: a standard CNN, the Slot Attention model, SA-MESH (Zhang et al., 2023), and CoSA (Kori et al., 2024a). Notably, CoSA is a recent work and, to the best of our knowledge, the only existing method that has explicitly attempted tasks of this type. Additionally, for tasks that do not require generalization to larger numbers of objects, we include comparisons with a standard neurosymbolic (NeSy) implementation following the setup described in Misino et al. (2022) for the classification task. Full details of the baseline configurations and implementations are provided in Appendix B.

**Metrics.** We report task accuracy, defined as the proportion of correctly classified images, and concept accuracy, which measures the correctness of the inferred object representations. Note that concept accuracy is not computable for fully neural baselines, as they do not model objects explicitly.

## 6 RESULTS

We evaluate DeepObjectLog's ability to generalize across the four core research questions introduced above. Results are summarized in Table 1 and Table 2.

**Q1: DeepObjectLog generalizes more robustly to novel combinations of familiar objects.** In the MM-A task (Table 1), our model achieves a Task Accuracy of 90% on out-of-distribution (OOD) compositions, substantially outperforming neural baselines and existing object-centric models. This demonstrates the model's ability to exploit the provided logic knowledge to learn more meaningful object representations that are more independent on the particular combinations seen in the training set. At the same time, this improvement necessitates an object-centric perceptual architecture: while NeSy shares the same symbolic reasoning component, it achieves only 36% in this setting.

**Q2: DeepObjectLog allows for generalization to novel classes.** In the PokerRules task (Table 2), our model achieves an OOD class accuracy of 72%, clearly outperforming NeSy (0.46%) and neural baselines, which cannot generalize to unseen classes under the same conditions. Similar results are also reported for the MM-A (see Appendix C) and the CLEVR-A tasks. This is because the model is not merely learning input–output associations but it better decomposes the input into *meaningful object-level representations*, as the Concept Accuracy results shows in Table 1. This object-centric decomposition makes the system both adaptable and interpretable: by grounding predictions in symbolic concepts, the same perceptual model *can be reused for different reasoning tasks on the same input domain* simply by modifying the logical program, with the resulting accuracy being at least as high as the model's concept accuracy.

**Q3: DeepObjectLog adapts to new structural configurations.** When evaluated on images containing a number of objects not seen during training, our model generalizes more effectively than neural baselines. In the MM-A task, the model maintains substantially higher accuracy on extrapolation test sets with four and five digits, while competitors show significant degradation. The interpolation results, reported in Appendix C, also show a similar trend. Similarly, in the PokerRules and CLEVR-A tasks, the model shows stronger extrapolation ability than competitors both on known classes and

Table 1: Task Accuracy (overall image) and Concept Accuracy (object-level decomposition) in MM-A. Results are on a test set of in-distribution compositions (**Test**) and on out-of-distribution compositions (**OOD**) . We also report extrapolation performance on images with four or five digits OOD counts. Dashes indicate cases where a model cannot provide results due to architectural limitations.

| MM-A | Task Acc. | | Concept Acc. | | Extrapolation | |
|---|---|---|---|---|---|---|
| | Test | OOD | Test | OOD | 4 digits | 5 digits |
| CNN | $79.90_{\pm0.17}$ | $3.43_{\pm0.15}$ | - | - | $22.16_{\pm0.90}$ | $13.16_{\pm0.96}$ |
| SA | $\mathbf{98.90}_{\pm0.34}$ | $13.30_{\pm3.14}$ | - | - | $36.23_{\pm4.44}$ | $9.76_{\pm6.11}$ |
| SA-MESH | $98.86_{\pm0.47}$ | $18.26_{\pm1.33}$ | - | - | $37.50_{\pm1.15}$ | $12.00_{\pm0.51}$ |
| CoSA | $93.00_{\pm1.92}$ | $36.2_{\pm15.10}$ | - | - | $52.20_{\pm14.01}$ | $25.06_{\pm12.82}$ |
| NeSy | $90.70_{\pm4.02}$ | $35.76_{\pm6.24}$ | $55.43_{\pm26.40}$ | $23.83_{\pm3.82}$ | - | - |
| **Ours** | $94.26_{\pm2.00}$ | $\mathbf{90.00}_{\pm3.01}$ | $\mathbf{85.16}_{\pm2.45}$ | $\mathbf{65.46}_{\pm5.70}$ | $\mathbf{69.73}_{\pm10.74}$ | $\mathbf{44.06}_{\pm3.85}$ |

Table 2: Task Accuracy for models evaluated on PokerRules and CLEVR-Addition (CLEVR-A), with results reported on both in-distribution test classes and OOD classes. A dash indicates cases where a model cannot provide results due to architectural limitations.

| | Test | OOD class | In-distribution class | OOD class |
|---|---|---|---|---|
| PokerRules | | | Extrapolation: 5 cards | |
| CNN | $81.96_{\pm1.10}$ | - | $22.03_{\pm5.27}$ | - |
| SA | $99.30_{\pm1.21}$ | - | $35.86_{\pm8.72}$ | - |
| SA-MESH | $\mathbf{99.93}_{\pm0.11}$ | - | $37.80_{\pm4.91}$ | - |
| CoSA | $95.46_{\pm3.90}$ | - | $44.26_{\pm17.16}$ | - |
| NeSy | $80.23_{\pm2.11}$ | $0.46_{\pm0.05}$ | - | - |
| **Ours** | $97.90_{\pm1.17}$ | $\mathbf{72.23}_{\pm16.72}$ | $\mathbf{78.53}_{\pm4.68}$ | $\mathbf{78.23}_{\pm19.24}$ |
| CLEVR-A | | | Extrapolation: 7 objects | |
| CoSA | $84.79_{\pm3.95}$ | | $3.44_{\pm1.69}$ | - |
| **Ours** | $\mathbf{93.12}_{\pm0.56}$ | | $\mathbf{59.81}_{\pm12.45}$ | $\mathbf{28.57}_{\pm3.71}$ |

previously unseen class configurations, where other approaches either fail to generalize or are not applicable. This result reflects the synergy between the model's ability to decompose the scene and the object-centric mechanism's inherent independence from the number of extracted objects.

**Q4: DeepObjectLog shows superiority over pretrained large Vision-Language Models.** We evaluated LLaVA-1.5-7B-hf and Qwen2.5-VL-7B-Instruct on MM-A by providing the task description and possible answers. Despite their scale ($\sim$7B parameters), both models struggled: the best competitor achieved only 59.93% accuracy on OOD compositions. In contrast, our approach ($\sim$2M parameters) reached 90% accuracy. While this comparison is not strictly fair, since these models are not explicitly trained for this task, it demonstrates that the benchmark is non-trivial even for strong general-purpose VLMs. Details and results are provided in Appendix C.

## 7 CONCLUSIONS

We propose DeepObjectLog, a neurosymbolic model that integrates object-centric perception with probabilistic logical reasoning to jointly infer and reason over a variable number of latent objects. Our results show that (1) the proposed method strongly outperforms neural and neurosymbolic baselines in compositional generalization tasks, (2) it effectively generalizes to scenes with previously unseen object counts and classes, and (3) it enables robust reasoning under uncertainty without requiring explicit object-level supervision. These advances mark an important step toward more general and adaptive neurosymbolic systems, contributing to the development of robust and interpretable AI.

**Limitations and future work.** Our approach shares several limitations with current object-centric and neurosymbolic methods. First, it is less scalable than purely neural approaches, despite mitigation

by Maene et al. (2024) (Appendix C). Second, evaluation on real-world datasets remains an open challenge for the entire field, as existing models still struggle in such settings (Seitzer et al., 2022). Finally, in scenarios where visual or label signals are insufficient, models may fall back on reasoning shortcuts (Marconato et al., 2023). We view our contribution as a step toward overcoming these broader challenges.

## 8 REPRODUCIBILITY STATEMENT

All necessary details to reproduce our experiments are provided in Appendix B, including a full description of the architectures and the datasets employed. This appendix contains hyperparameters, training settings, and implementation notes to ensure faithful replication of our results. The source code will be released publicly upon acceptance.

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

# Supplementary Material

T­ABLE OF C­ONTENTS

## A  OBJECTIVE DERIVATION

Consider the marginal:

$$p(y,x) = \int dz\, p(x,y,z) = \int dz\, p(z)p(x|\boldsymbol{s}^*(z), \boldsymbol{\beta}^*(z))p(y|\boldsymbol{s}^*(z))$$

Consider the logarithm:

$$\log p(x,y) \;=\; \log \int p(x,y,z)\,dz \;=\; \log \int \underbrace{\exp\big(f(z)\big)}_{f(z)\,:=\,\log p(x,y,z)} dz \tag{6}$$

Using the log-sum-exp bound we obtain:

$$\int \exp(f(z))dz \;\geq\; \max_z \exp(f(z)) = \exp\big(\max_z f(z)\big),$$

Therefore:

$$\log p(x,y) \;\geq\; \max_z \log p(x,y,z) \;=\; \log p\big(x,y,z^\star\big)),$$

where $z^\star := \arg\max_z \log p_\theta(x,y,z)$ is the maximum-a-posteriori (MAP) estimate of $z$.

Optimising $z^\star$ separately for every training case is expensive, so we amortize it with a neural encoder $z = q(x;\phi)$ Substituting $q(x) = q(x;\phi)$ for $z^\star$, we obtain:

$$\log p(x,y) \;\geq\; \log p\big(x,y,q(x)\big). \tag{7}$$

which is a *lower bound* on the true marginal log-likelihood. Maximising it with respect to all parameters $\{\theta,\phi\}$ therefore pushes the exact marginal likelihood upwards, while avoiding any intractable integral.

## B  EXPERIMENTAL DETAILS

### B.1  ARCHITECTURAL DETAILS

All architectures and code used in this study were implemented in Python using PyTorch, with DeepProbLog employed for the symbolic integration. We provide here the architectural details of the baseline models and our proposed method used in the experiments.

**CNN**    The CNN baseline consists of a convolutional encoder followed by a multi-layer perceptron (MLP) classifier over the target classes. The encoder comprises a sequence of convolutional layers with kernel size 5, each followed by a ReLU activation. The first convolutional layer uses a stride of 2 to reduce spatial resolution, while the remaining layers use a stride of 1. The output of the final convolutional layer is processed by an Adaptive Average Pooling layer. The resulting feature vector is then passed to an MLP with hidden dimension 64, which outputs the class logits.

**Slot Attention**    For Slot Attention, we adopt the architecture from the original paper. Both the slot dimension and the embedding dimension are set to 64, and the number of slot attention iterations is 2. For the final classification, we apply an MLP over the summed slot embeddings. The decoder module follows the design proposed in the original Slot Attention paper and several PyTorch implementations are available online.

**SA-MESH**    The SA-MESH baseline builds on the Slot Attention architecture, with the addition of mesh-based interactions (Zhang et al., 2023). Both the slot dimension and the embedding dimension are set to 64, and the number of slot attention iterations is 2. The remaining hyperparameters match those in the original SA-MESH repository. For the final classification, as in Slot Attention, we apply an MLP over the summed slot embeddings. The decoder employed matches that used in the Slot Attention baseline.

**CoSA**    The CoSA implementation code is provided by the authors of the paper and is available online, even if not all hyperparameters of the model were explicitly provided in the released code. Based on reported results, we selected the Gumbel variant with 3 slot attention iterations as the best-performing configuration and used it across all experiments. For the MM-A and Poker Rules tasks we adapted the reasoning classifier described in their work to fit the specific tasks considered in our evaluation (i.e. setting the correct amount of properties and classes).

This model required structural modifications for the CLEVR-A task due to the two input images used to calculate the sum. To adapt the model, we used the Slot Attention mechanism on each image separately and added the resulting slots to feed everything into a single classification head. As the reasoning classifier described in their paper cannot be used for this task without modification, we replaced it with a simple MLP classifier.

**NeSy baseline**    Following the idea exposed in Misino et al. (2022), in particular for the Label Classification task, the NeSy baseline uses a single CNN encoder backbone to process the input and leverages the DeepProbLog interface, mirroring our own method for fair comparison. To enable classification over multiple entities, the architecture includes two MLPs per possible object in the training set—one for estimating objectness and one for classification. All hyperparameters of the encoder align with those used in the CNN baseline. The symbolic part is interfaced through DeepProbLog, where each output of the objectness and classifiers represents a neural predicate. These predicates are then utilised in combination to construct the rules employed for the final classification of the task. An example pseudo-code for the MultiMNIST-addition task is available in Table 3.

**Our model**    Our proposed method was implemented using both the standard Slot Attention mechanism and the SA-MESH variant. As the latter showed slightly better average performance, we report results for the SA-MESH-based implementation. The hyperparameters follow those recommended in the SA-MESH paper, where maintaining a low number of slot iterations (2) and increasing the number of mesh iterations (5) proved beneficial. Unlike the NeSy baseline, our method employs only two MLPs in total (one for objectness and one for classification), which are applied separately to each extracted object representation. To condition both the class classifier and the reconstruction, we follow the Zero Slot Strategy from Adaptive Slot Attention: the objectness probability is multiplied with the slot before it is passed to the classifier and decoder (See the next subsection for details.). The decoder architecture is the same as used in Slot Attention and the symbolic interface is the same as described in the NeSy baseline.

As with CoSA, this model required some structural modifications for the CLEVR-A task due to the two input images. We simply adapted the model by using the Slot Attention mechanism separately on the two images. All classification results were then fed into the same logic circuit. Please note that, for this specific task, adding a classification head for the object properties is unnecessary, as the objectness head is sufficient.

**Soft masking**    Our implementation of soft masking draws inspiration from the Zero Slot strategy introduced in Adaptive Slot Attention (Fan et al., 2024). In their formulation, each slot representation $s_i$ is modulated by its associated objectness score (our $\beta_i$):

$$\tilde{s}_i = s_i \cdot \beta_i$$

However, unlike their approach, we do not apply a Gumbel-Softmax to discretize the objectness scores.

### B.2    TRAINING DETAILS

We run all our experiments on a Nvidia L40S 48GB GPU card with AMD EPYC 9334 CPU and 256GB RAM.

We train all models using the AdamW optimizer with a learning rate of 1e-4 and weight decay of 1e-4. Neural baselines are trained for 100 epochs with a batch size of 64. Neurosymbolic models are trained longer, depending on the dataset: 400 epochs with batch size 64 for MultiMNIST and PokerRules, and 200 epochs with batch size 32 for CLEVR-Addition (the same batch size is also used for CoSA on CLEVR-Addition). In all cases, the best model is selected based on validation performance. All experiments are run with multiple seeds, and we report mean and standard deviation across runs in the tables.

### B.3 DATASETS

All the data we used to build our datasets are freely available on the web with licenses:

- MNIST - CC BY-SA 3.0 DEED,
- Cards Image Dataset - CC0: Public Domain.
- CLEVR - CC BY 4.0

The original datasets used as a basis for constructing our synthetic benchmarks are not redistributed. We will make the generation code of our datasets available for reproducible experiments.

**MultiMNIST.** We build upon an existing MultiMNIST generator (Sun, 2019), modifying it to better suit the requirements of our experimental setup. In particular, we adapted the generation process to support distant supervision, where the model is trained only on the sum of the digits present in each image. The training set includes 30,000 images, and the validation and test set both include 3,000 images. Digit appearances are balanced to ensure uniform distribution across the dataset and we make sure that all possible output sums (generable from possible compositions) are seen during training. All images are normalized to the range $[-1, 1]$. While the models are trained only on the total sum of the digits, we retain the individual digit annotations to support the evaluation of concept-level accuracy.

Table 3: Example of predicates and background knowledge for MultiMNIST. This is a simplification of the version employed in the experiments, wherein a maximum of two digits are considered. It is important to note the role of $isobj\_tmp$ (objectness) in this process, as it facilitates the selection of elements to be added, thereby determining the final result, and those to be excluded.

$$
\begin{aligned}
&\%\ \textit{Neural\ predicates}\\
&classifier\_slot0(X, N) :: digit\_tmp(X, 0, N) : -between(0, 9, N).\\
&classifier\_slot1(X, N) :: digit\_tmp(X, 1, N) : -between(0, 9, N).\\
&isobject\_slot0(X) :: isobj\_tmp(X, 0).\\
&isobject\_slot1(X) :: isobj\_tmp(X, 1).\\
\\
&\%\ \textit{Conditioned\ digit\ definition}\\
&digit(X, ID, Y) : - isobj\_tmp(X, ID),\ digit\_tmp(X, ID, Y).\\
\\
&\%\ \textit{Rules}\\
&addit(X, ID0, SumIn, SumOut) : - isobj\_tmp(X, ID0),\ digit(X, ID0, C),\\
&\qquad\qquad\qquad\qquad\qquad\qquad SumOut\ is\ SumIn + C.\\
&addit(X, ID0, ID1, SumIn, SumOut) : - isobj\_tmp(X, ID1),\ digit(X, ID1, C),\\
&\qquad\qquad\qquad\qquad\qquad\qquad Y\ is\ SumIn + C,\ addit(X, ID0, Y, SumOut).\\
&addit(X, ID0, SumIn, SumOut) : - not(isobj\_tmp(X, ID0)).\\
&addit(X, ID0, ID1, SumIn, SumOut) : - not(isobj\_tmp(X, ID1)),\\
&\qquad\qquad\qquad\qquad\qquad\qquad addit(X, ID0, SumIn, SumOut).\\
\\
&addition(X, Z) : -addit(X, 0, 1, 0, Z).\\
\\
&\%\ \textit{Query}\\
&? - addition(input, Z)).
\end{aligned}
$$

**PokerRules.** The PokerRules dataset is synthetically generated to support relational reasoning from visual input, where the label depends on the combination of card types present in the scene. The training set includes 20,000 images, with 2,000 images each for validation and testing. Each image contains between 1 and 4 playing cards (5 for the extrapolation setting) positioned along a horizontal line with slight random perturbations to their location. The cards are drawn from a fixed suit (hearts) and include the ranks: ten, jack, queen, king, and ace. Images are normalized to the range $[-1, 1]$. The training set includes five classes: nothing (1–4 cards), pair (2–4 cards), two pairs, poker and straight (4 cards). The appearance of each class is balanced across training samples. To evaluate the model's ability to generalize, we introduce three test sets. In the first, we add the unseen class *three of a kind*, an out-of-distribution composition where three identical cards appear together (with 3 or 4 cards in total). In the second, we test *poker-5*, where a standard poker hand is accompanied by an additional unrelated card. This introduces extrapolation to a higher number of objects (five), while keeping the underlying class structure familiar. Finally, in the third test set we evaluate *full house*, a completely novel hand type composed of three cards of one rank and two of another.

**CLEVR.** Starting from the CLEVR dataset (Johnson et al., 2017), we construct a new dataset tailored for the CLEVR-Addition task. Images are paired randomly, and each pair is assigned as label the sum of the number of objects contained in the two images. All images are normalized to the range $[-1, 1]$. For training and validation, we use the CLEVR training split, restricting to images with at most six objects. This results in 26,132 images, with the last 10,000 reserved for validation. All test sets are derived from the CLEVR validation split to prevent data leakage. We construct three test sets:

- In-distribution: pairs of images containing at most six objects, following the same protocol as for training.

- Extrapolation (same class space): pairs including images with seven objects, but only those combinations whose total sum remains within the training range (i.e., $\leq 12$). This ensures neural baselines can still attempt the task.

- Extrapolation + OOD classes: pairs combining images with six and seven objects, producing sums beyond the training range and thus introducing new, unseen classes.

# C   ADDITIONAL RESULTS

## C.1   COMPLEMENTARY EXPERIMENTS

This section presents complementary experiments on the MultiMNIST, PokerRules and CLEVR datasets.

**Comparison with foundation models.** To better understand how foundation models compare in our setting, we conducted an experiment using two famous Vision-Language Models: "LLaVA-1.5-7B-hf" and "Qwen2.5-VL-7B-Instruct", on our MultiMNIST-Addition task including only out-of-distribution test set for fair comparison with our model. The results of the experiment are available in Table 4. The VLMs were prompted with natural language queries describing the task and the possible solutions (i.e. how many digits can be in the image). Despite their general capabilities, we observe that they struggled to reliably solve the task.

Table 4: Accuracy results of our model and two Vision-Language Models on the MultiMNIST-addition task.

| Model | OOD Compositions | Extrap. (4 digits) | Extrap. (5 digits) |
|---|---|---|---|
| Llava-1.5-7b-hf | 27.43 | 2.73 | 1.87 |
| Qwen2.5-VL-7B-Instruct | 59.93 | 28.67 | 14.37 |
| **Ours** | **90.00** | **69.73** | **44.06** |

All experiments were conducted under controlled and fixed settings (e.g., decoding temperature, and evaluation protocols) to ensure full reproducibility of results.

The prompts used for the experiments for each of the three test sets are reported below.

- **OOD compositions**: *"What is the value of the sum of the digits in this picture? An image with only a black background is just a 0. Reply only with the sum value. Note that the sum values can be between 0 and 27 and the image contains between 0 and 3 digits."*

- **Extrapolation (4 digits)**: *"What is the value of the sum of the digits in this picture? An image with only a black background is just a 0. Reply only with the sum value. Note that the sum values can be between 0 and 36 and the image contains between 0 and 4 digits."*

- **Extrapolation (5 digits)**: *"What is the value of the sum of the digits in this picture? An image with only a black background is just a 0. Reply only with the sum value. Note that the sum values can be between 0 and 45 and the image contains between 0 and 5 digits."*

**Interpolation.** In Table 5, we report results from the interpolation setting on MultiMNIST, where models are trained on images containing 0, 1, or 3 digits and tested on samples containing exactly 2 digits. As with other setups, our model demonstrates superior generalization compared to all baselines.

Note that, unlike the extrapolation setting, the NeSy baseline is applicable here and provides out-of-distribution predictions that are better than the neural baselines. Additionally, the interpolation setup is particularly challenging, as the model is exposed to fewer digit configurations overall (only three instead of four, as in the standard setting), which increases the difficulty of identifying structural patterns. This suggests that providing a broader range of digit counts during training (e.g., including 4 or 5 digits) could further enhance interpolation performance.

Table 5: Accuracy results on the MultiMNIST-addition interpolation setting.

| Model | Test (0,1,or 3 digits) | Interpolation (2 digits) |
|---|---|---|
| CNN | $76.50_{\pm 0.70}$ | $24.90_{\pm 0.14}$ |
| SA | $98.40_{\pm 0.70}$ | $10.05_{\pm 3.74}$ |
| SA-MESH | $\mathbf{99.05}_{\pm 0.07}$ | $14.60_{\pm 5.09}$ |
| CoSA | $87.23_{\pm 5.85}$ | $22.53_{\pm 20.16}$ |
| NeSy | $94.35_{\pm 0.49}$ | $33.25_{\pm 7.42}$ |
| **Ours** | $89.80_{\pm 6.34}$ | $\mathbf{62.30}_{\pm 19.44}$ |

**Supplementary results for task generalization.** Table 6 presents results in an extrapolation setting involving both seen and unseen classes. The rightmost columns focus on unseen classes, where the model is tested on images containing 4 or 5 digits with sum values beyond those encountered during training. This scenario highlights the adaptability of our approach: none of the baselines can be applied here, as they lack either structural or task generalization capabilities. Our model, instead, handles both simultaneously.

Table 6: Accuracy results on the MultiMNIST-addition extrapolation settings. The right columns show the case where both the class and the number of digits are out of distribution.

| | Extrapolation | | OOD class extrapolation | |
|---|---|---|---|---|
| | 4 digits | 5 digits | 4 digits | 5 digits |
| CNN | $22.16_{\pm 0.90}$ | $13.16_{\pm 0.96}$ | - | - |
| SA | $36.23_{\pm 4.44}$ | $9.76_{\pm 6.11}$ | - | - |
| SA-MESH | $37.50_{\pm 1.15}$ | $12.00_{\pm 0.51}$ | - | - |
| CoSA | $52.20_{\pm 14.01}$ | $25.06_{\pm 12.82}$ | - | - |
| **Ours** | $\mathbf{69.73}_{\pm 10.74}$ | $\mathbf{44.06}_{\pm 3.85}$ | $\mathbf{70.50}_{\pm 9.46}$ | $\mathbf{46.66}_{\pm 5.14}$ |

**Concept accuracy.** In Table 7, we provide additional metrics for the PokerRules task, including concept accuracy and card-count prediction accuracy on both the in-distribution test set and the test set with an out-of-distribution class. Unlike MultiMNIST, this task lacks an inductive bias strong enough to help the model identify individual cards using only distant supervision. Still, our model achieves strong accuracy in estimating the number of cards per image. We also report a best-matching concept accuracy, computed by mapping the predicted card classes to the most plausible configuration of symbols. The results reflect the increased complexity of this task, yet our model still outperforms the neurosymbolic baseline.

We also note that we tested our model's concept accuracy in the CLEVR-Addition task, achieving an excellent score of $96.28\%_{\pm 0.55}$. Here, the model, starting from the addition value, was able to understand the number of objects present in a single image.

Table 7: Concept accuracy results for PokerRules

| | Task accuracy | | Concept accuracy | | Objects number accuracy | |
|---|---|---|---|---|---|---|
| | Test | OOD class | Test | OOD class | Test | OOD class |
| NeSy | $80.23_{\pm 2.11}$ | $0.46_{\pm 0.05}$ | $18.63_{\pm 1.37}$ | $0.13_{\pm 0.05}$ | $80.20_{\pm 5.82}$ | $76.53_{\pm 24.88}$ |
| **Ours** | $\mathbf{97.90}_{\pm 1.17}$ | $\mathbf{72.23}_{\pm 16.72}$ | $\mathbf{32.00}_{\pm 5.88}$ | $\mathbf{29.26}_{\pm 13.95}$ | $\mathbf{81.60}_{\pm 2.78}$ | $\mathbf{84.13}_{\pm 6.60}$ |

**Ablation study.** In Table 8, we present an ablation study on the MultiMNIST-Addition task to assess the contribution of key components in our architecture. Each variant highlights the role of architectural choices in supporting accurate object-level reasoning and task performance.

Table 8: Ablation results on the MultiMNIST-Addition task. Specifically, we evaluate performance under the following modifications: (1) replacing the object-centric encoder SA-MESH with the standard Slot Attention mechanism; (2) using a single classifier that treats non-objectness as an additional class (common approach on object-centric learning), instead of our two-head design with separate objectness and class classifiers; (3) removing the conditioning of the class classifier on the objectness score; (4) excluding the final decoding phase and corresponding reconstruction loss during training. Regarding this last point, a recent work (Marconato et al., 2023) has shown that incorporating perceptual cues through reconstruction can help align perception with symbolic reasoning, discouraging reliance on irrelevant features and preventing to fall in reasoning shortcuts.

|  | Task Accuracy | | Concept Accuracy | |
|---|---|---|---|---|
|  | Test | OOD comps | Test | OOD comps |
| (1) Original SA | $76.91_{\pm 3.25}$ | $71.70_{\pm 3.53}$ | $30.70_{\pm 5.70}$ | $5.71_{\pm 0.14}$ |
| (2) One classifier only | $93.49_{\pm 0.70}$ | $64.06_{\pm 19.09}$ | $15.05_{\pm 9.19}$ | $20.49_{\pm 14.14}$ |
| (3) No class conditioning | $92.51_{\pm 0.28}$ | $81.84_{\pm 3.46}$ | $66.77_{\pm 14.96}$ | $41.16_{\pm 12.10}$ |
| (4) No reconstruction | $\mathbf{95.75}_{\pm 1.76}$ | $89.55_{\pm 2.19}$ | $64.64_{\pm 2.33}$ | $52.20_{\pm 11.59}$ |
| **Ours** | $94.26_{\pm 2.00}$ | $\mathbf{90.00}_{\pm 3.01}$ | $\mathbf{85.16}_{\pm 2.45}$ | $\mathbf{65.46}_{\pm 5.70}$ |

## C.2 TRAINING PERFORMANCE

The processing speed in terms of iteration/sec for our model and the baselines is tabulated in Table 9. It is to be noted that speed might differ slightly with respect to the considered system and the background processes.

Table 10 reports the total training time for all models in the main experiments, measured with a single seed. The project required additional computing power when testing different variations of our model and hyperparameter tuning.

Table 9: Training speed (iterations per second, it/s) of the evaluated models across the three experimental settings.

| Methods (↓), Batch size (→) | MM-A 64 | PokerRules 64 | CLEVR-A 32 |
|---|---|---|---|
| CNN | 24.60 | 6.97 | N.A. |
| SA | 14.06 | 3.72 | N.A. |
| SA-MESH | 11.06 | 3.12 | N.A. |
| CoSA | 6.76 | 3.65 | 2.15 |
| NeSy | 12.43 | 3.23 | N.A. |
| Ours | 7.66 | 2.1 | 1.75 |

Table 10: Total training time per epoch (hours:minutes) for the evaluated models across the three experimental settings.

| Methods (↓), | MM-A | PokerRules | CLEVR-A |
|---|---|---|---|
| CNN | ∼0h 31min | ∼1h 14min | N.A. |
| SA | ∼0h 55min | ∼2h 20min | N.A. |
| SA-MESH | ∼1h 10min | ∼2h 47min | N.A. |
| CoSA | ∼1h 55min | ∼2h 22min | ∼5h 03min |
| NeSy | ∼4h 11min | ∼10h 46min | N.A. |
| Ours | ∼6h 47min | ∼16h 33min | ∼9h 20min |

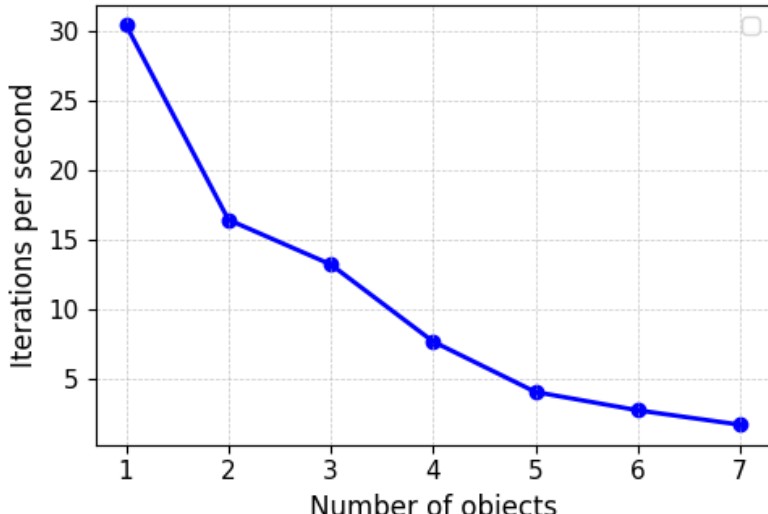

Figure 3: Number of inference iterations per second for the MM-A task (batch size 64) as the number of objects increases.

### C.3 ATTENTION WEIGHTS ANALYSIS

We present the attention weights produced by our neurosymbolic model and by SA-MESH on discriminative tasks trained with distant supervision. These weights are obtained from the last iteration of the slot attention module and visualized as spatial maps over the input. The comparison includes in-distribution, out-of-distribution, and extrapolation settings. Our model's attention masks clearly show its ability to separate and individually attend to the objects in the scene. In contrast, the purely neural SA-MESH model appears less capable of capturing the underlying compositional structure of the image, often focusing on the overall configuration rather than the individual objects relevant to solving the task.

Analyzing the correspondence between attention weights and the predicted objectness and classification values for each slot, we observe that slots attending to multiple regions simultaneously tend to receive low objectness scores, whereas slots with sharp, localized attention typically exhibit high objectness confidence.

In-distribution

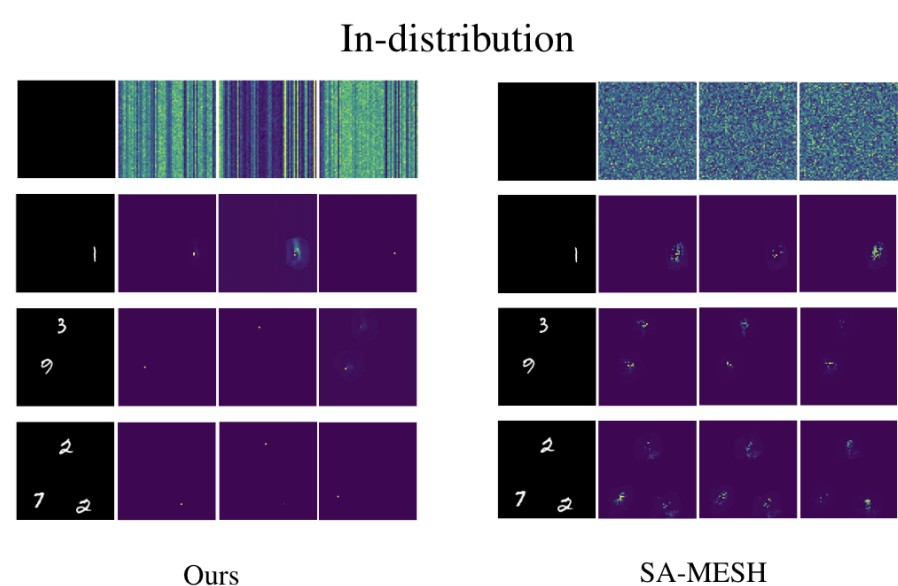

Ours                                     SA-MESH

Figure 4: Attention weights of the Slot Attention mechanism for the MultiMNIST dataset.

Extrapolation

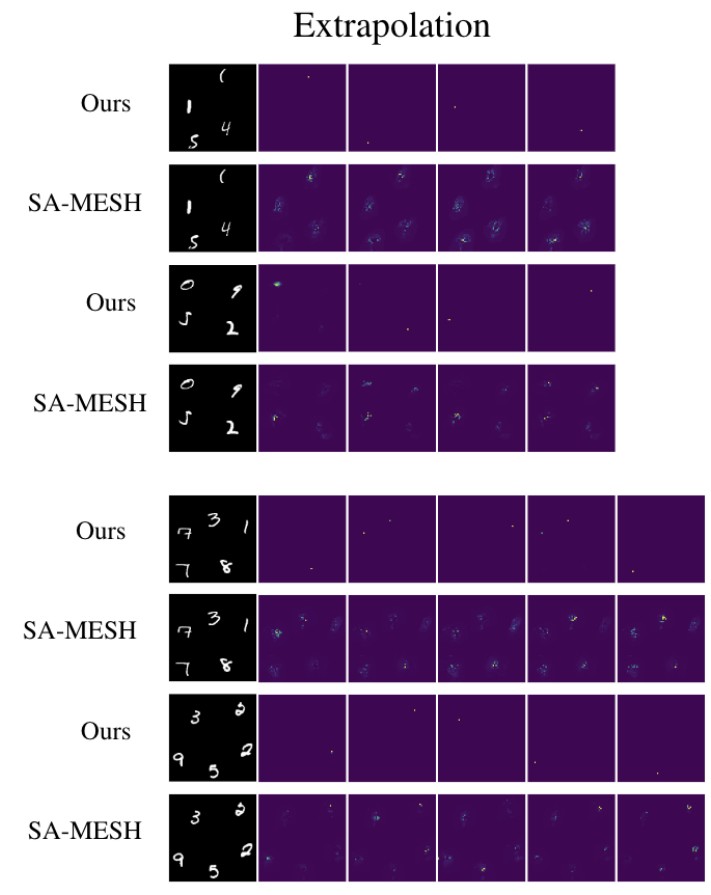

Figure 5: Attention weights of the Slot Attention mechanism for the MultiMNIST dataset.

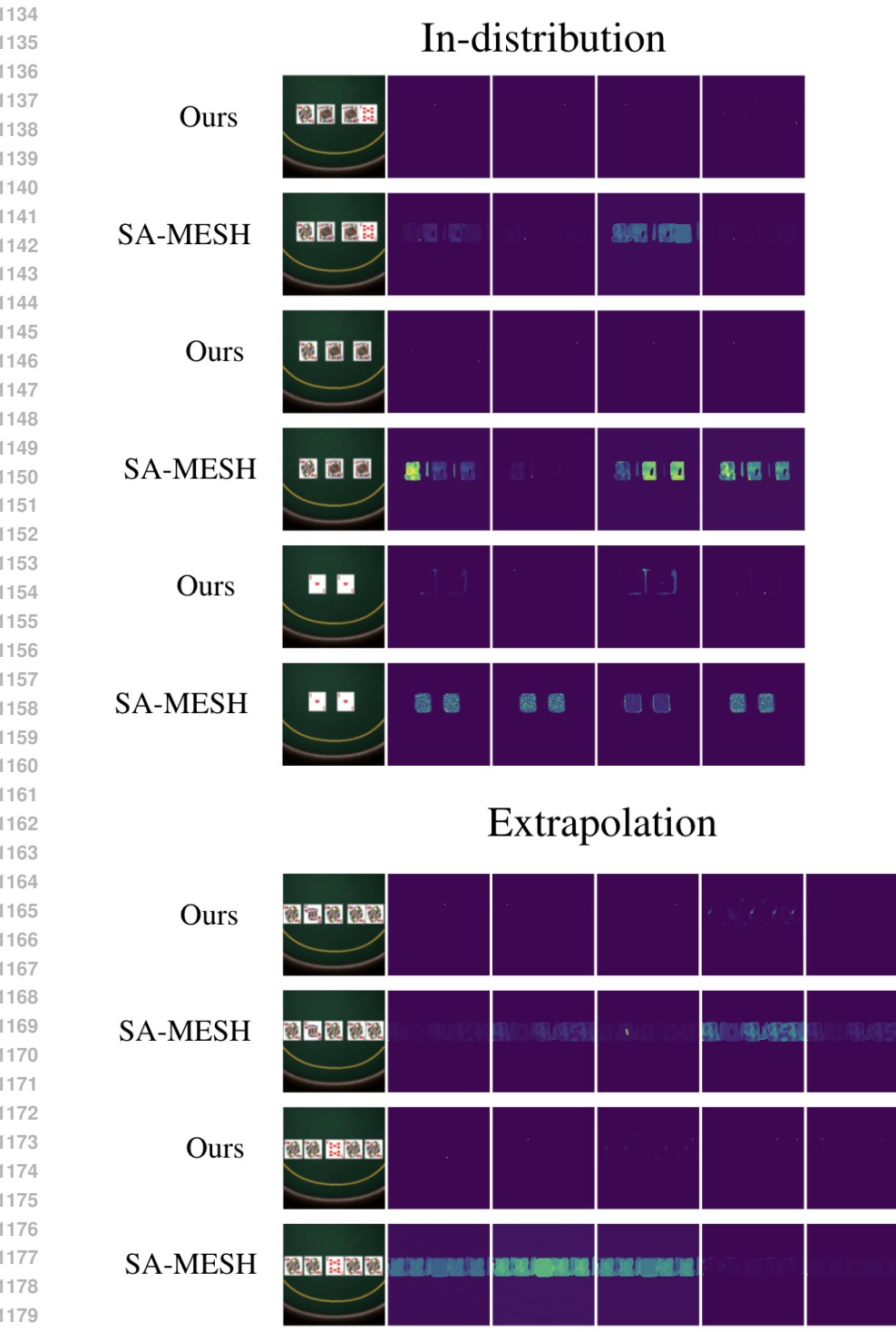

Figure 6: Attention weights of the Slot Attention mechanism for the PokerRules dataset. As the images are larger than those in MultiMNIST, it is harder to see where the attention weights are highest. However, the model is exhibiting the same behaviour as in previous experiments.

## C.4 ALPHA MASKS ANALYSIS

In this section, we present examples of the alpha masks produced by our model during the reconstruction phase. These masks are primarily useful for reconstruction, while the attention weights are instead used to compute and refine the slot representations. Notably, the attention weights often focus on small regions within objects, whereas the alpha masks tend to approximate the full segmentation of the objects (Zhang et al., 2023). For each alpha mask, we also report the corresponding objectness predicted by the model for that slot. The associated score is directly used in the reconstruction phase.

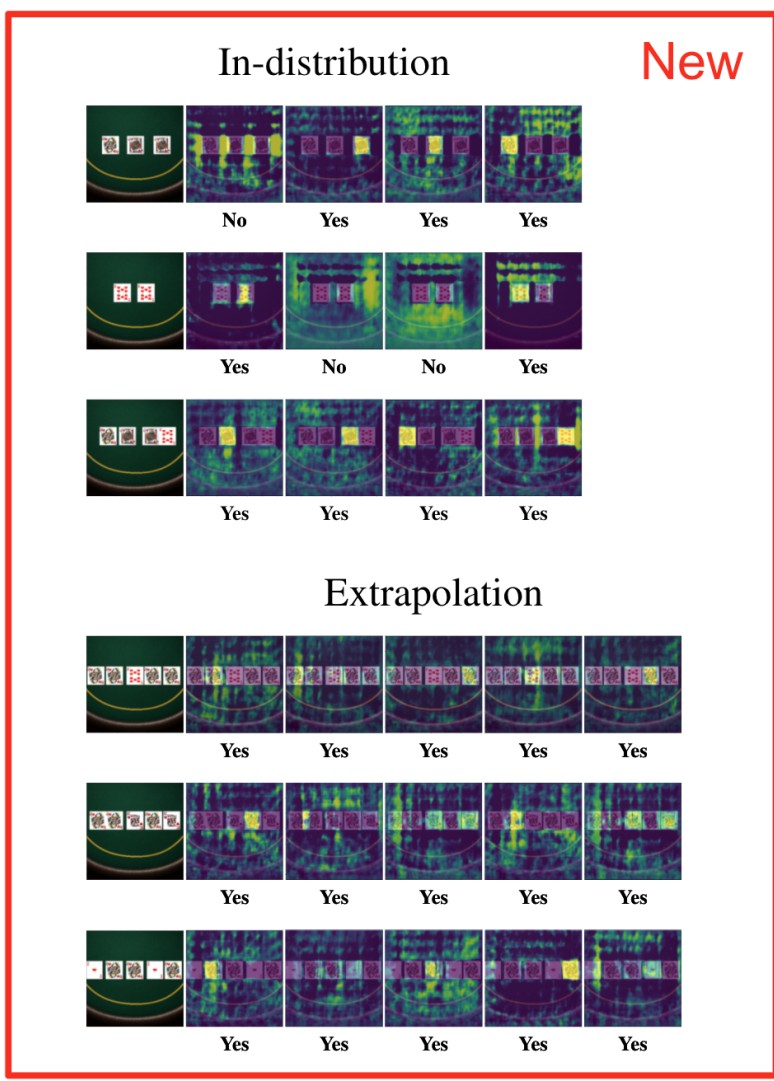

Figure 7: Alpha masks produced by the Slot Attention mechanism trained with our method on the PokerRules dataset. For each slot, the associated objectness score is shown, with slots above 0.5 marked as *yes*.

## D    LLMs USAGE DECLARATION

The authors declare that, in the writing process, Large Language Models were used exclusively to polish and improve the clarity of the text.

