# OpenReview forum: "Neurosymbolic Object-Centric Learning with Distant Supervision"
_ICLR.cc/2026/Conference — Submitted to ICLR 2026_

### Official Review · Reviewer_Nexu · 2025-10-29

**Soundness:** 3
**Presentation:** 3
**Contribution:** 3
**Rating:** 6
**Confidence:** 2

**Summary:**

This paper addresses the challenge of learning relational reasoning directly from raw perceptual data without explicit object-level supervision. The authors aim to bridge the gap between perception (inferring objects from images) and reasoning (logical inference over those objects) in order to achieve systematic generalization beyond the training distribution. In particular, they consider tasks where supervision is only given as high-level labels (distant supervision), such as the sum of digits in an image or the category of a hand of cards, rather than any annotations of individual objects.

Main Contributions: (1) The paper highlights limitations of prior neural and neurosymbolic models, which either required object-level supervision or assumed a fixed, known decomposition of the input. It proposes a new probabilistic generative formulation that learns object-centric representations directly from raw images with only task-level (distant) supervision. (2) It instantiates this formulation in the DeepObjectLog architecture, combining slot-based neural perception with probabilistic logic programming (ProbLog) for end-to-end differentiable reasoning. This design satisfies key requirements: handling a variable number of objects, reasoning under uncertainty, and inducing relational structure without direct object labels. (3) The authors introduce new benchmark tasks to evaluate relational generalization under weak supervision, including a MultiMNIST addition task (sum of digits in an image), a CLEVR-Addition task (adding counts of objects in paired images), and a newly created PokerRules dataset (classifying poker hands from images of playing cards). These tasks are specifically designed to test generalization to unseen object combinations, unseen object counts, and even novel task rules or classes not encountered in training. (4) Comprehensive experiments show that DeepObjectLog consistently outperforms baseline methods – both purely neural models and prior neurosymbolic approaches – especially in out-of-distribution generalization settings. The model is able to generalize to novel combinations of familiar objects, new object categories, and larger numbers of objects more robustly than the baselines, while also providing interpretable object-level predictions.

**Strengths:**

1.  Innovative Neurosymbolic Integration: The paper successfully combines slot-based object-centric perception with probabilistic logic programming. This is a novel integration – prior works did not achieve end-to-end training of object discovery guided by logical reasoning.
2. Effective Use of Distant Supervision: DeepObjectLog learns from only high-level task labels (distant supervision) and avoids any direct supervision on object existence or identity. This weakly-supervised paradigm is very appealing – it means the model can be trained on data where only aggregate or logical labels are provided.
3. Strong Generalization Performance: A major strength is the model’s performance on out-of-distribution generalization tests. DeepObjectLog dramatically outperforms baseline models in all the evaluated scenarios.

**Weaknesses:**

1. Dependence on Provided Logical Rules: A potential weakness is that the approach requires expert-defined logical rules for each new task. The logic component is not learned from data; it must be coded (in ProbLog) to describe the task’s outcome in terms of object properties.
2. Scalability and Complexity: The integration with probabilistic logic programming can introduce computational overhead. ProbLog, while efficient for moderate-scale inference, may struggle as the number of objects or logical facts grows large. The paper’s experiments involve at most 4–6 objects (cards or digits) and relatively simple rules. It is unclear how well the method scales to scenes with, say, dozens of objects or highly complex rule sets.
3. Synthetic Data and Visual Complexity: All evaluation tasks are on synthetic or simplistic visual data – e.g., MNIST digits on blank backgrounds, rendered CLEVR objects, and playing card images on a plain background. While this is suitable for controlled experiments, it leaves open the question of how the method handles more complex real-world images (with clutter, occlusion, varying backgrounds, etc.).
4. Hyperparameter Sensitivity: The approach introduces a few new hyperparameters – notably the maximum number of slots $N$.

**Questions:**

1. In the PokerRules experiment, the model was tested on an unseen hand type (“three of a kind”) and achieved 72% accuracy OOD. Could you clarify how this was handled in terms of the logic program and training? Specifically, was the logical rule for detecting a “three of a kind” included in the model from the start, or introduced only at test time? If the rule was added at test time, it’s impressive that the model adapts; however, how was the model’s output space configured to allow a new class it never saw during training (did the output simply come from the logic query without needing a dedicated neural output)? A detailed explanation of how novel classes are introduced and whether any fine-tuning is required would be helpful.
2. How does the training and inference time scale with the number of slots (objects) and the complexity of the logic program? For instance, if we were to increase $N$ or add more complex rules (say involving all pairs of objects or higher-order relations), does the inference via ProbLog become a bottleneck? The current tasks have relatively small $N$ (up to 5 objects). It would be useful if you could comment on any experiments or observations about performance as $N$ grows. Additionally, do you use any optimizations for the logic module (e.g., caching of inference results, using the structure of the logic to prune possibilities) during learning? Understanding the computational limits will inform how far this approach might scale to more complex scenes.
3. To disentangle where the gains come from, did you consider a variant of the model that does not use the logical loss but still uses the slot-based architecture (for example, training the slots + classifier purely on predicting the global label, akin to a black-box neural network without logic)?

---

> ### Author Response · Authors · 2025-11-21
>
> We thank the reviewer for highlighting both the strengths of our approach and the relevance of the generalization results. We appreciate the constructive comments on limitations and open questions, and we address each point below with clarifications, corrections, and additional evidence where helpful.
>
> &nbsp;
>
> **Question 1: In the PokerRules experiment, the model was tested on an unseen hand type (“three of a kind”) and achieved 72% accuracy OOD. Could you clarify how this was handled in terms of the logic program and training?
> A detailed explanation of how novel classes are introduced and whether any fine-tuning is required would be helpful.**
>
> **The rule detecting Three of a Kind was introduced only at test time, without any retraining or fine-tuning.** This is a direct consequence of how DeepObjectLog is structured: the symbolic program can be extended or modified without touching the perceptual module or the neural–symbolic interface. When a new rule is added, the neural component continues to output probabilistic slot-level predictions, and the logic engine simply evaluates this new rule on top of the existing interface.
> **This design gives our model a form of task flexibility that purely neural systems do not naturally support.** In many neural baselines, changing the reasoning objective (e.g., switching from one aggregation rule to another) requires collecting new labels and fine-tuning the model, and often still suffer from performance degradation (as documented in, e.g., Kori et al.). In DeepObjectLog, by contrast, the perceptual and symbolic components are cleanly separated: once the model has learned to extract object-centric concepts, adapting to a new reasoning structure amounts to supplying a different symbolic program.
>
> &nbsp;
>
> **Question 2: How does the training and inference time scales with the number of slots (objects) and the complexity of the logic program? The current tasks have relatively small N (up to 5 objects). Additionally, do you use any optimizations for the logic module during learning?**
>
> We acknowledge that probabilistic inference can become a bottleneck as the number of objects or classes grows, a challenge shared by all neurosymbolic systems and an active area of research. While our work does not propose new methods for scalable symbolic inference, we are already using recent state-of-the-art exact inference backends (e.g., based on Maene et al.,  KLay: Accelerating Arithmetic Circuits for Neurosymbolic AI [1]) to improve performance. Moreover, our system could benefit from approximate inference techniques, which are fully compatible with our model [2,3,4].
>
> Empirically, the method already handles scenes with substantially more than five objects. For example, in the CLEVR-Addition extrapolation setting it performs reasoning over **up to 14 objects per sample** (7 per image, since the task compares two images). **We will also include in the appendix a plot showing how inference time scales with the number of objects (figure 3)**, to allow a clearer understanding of the model’s scaling behaviour.
>
> **References:**
>
> [1] Maene, Jaron, Vincent Derkinderen, and Pedro Zuidberg Dos Martires. "KLay: Accelerating Arithmetic Circuits for Neurosymbolic AI." The Thirteenth International Conference on Learning Representations.
>
> [2] De Smet, Lennert, et al. "Relational neurosymbolic Markov models." Proceedings of the AAAI Conference on Artificial Intelligence. Vol. 39. No. 15. 2025.
>
> [3] van Krieken, Emile, et al. "A-nesi: A scalable approximate method for probabilistic neurosymbolic inference." Advances in Neural Information Processing Systems 36 (2023)
>
> [4] De Smet, Lennert, Emanuele Sansone, and Pedro Zuidberg Dos Martires. "Differentiable sampling of categorical distributions using the catlog-derivative trick." Advances in Neural Information Processing Systems 36 (2023).

---

> ### Author Response · Authors · 2025-11-21
>
> **Question 3: To disentangle where the gains come from, did you consider a variant of the model that does not use the logical loss but still uses the slot-based architecture?**
>
> **This comparison is already included in our experimental design.** Our baselines were selected precisely to disentangle the contribution of the symbolic component from that of the object-centric encoder. We evaluate DeepObjectLog against purely neural object-centric models trained without any logical loss, including standard Slot Attention and CoSA (ICLR 2024), whose neural reasoning module provides a black-box alternative to symbolic inference. For example, Slot Attention is simply DeepObjectLog without the logical layer, actually behaving as an ablated version of our model.
> These baselines allow us to assess how far a slot-based architecture can go when trained solely to predict the global label. Across all datasets, the results reported in the main paper already show a clear and consistent advantage for our logic-guided model.
> To make the source of these gains even more explicit, Appendix C (Table 8) presented an ablation study that isolates the impact of individual components of our architecture.
>
> &nbsp;
>
> **Weakness 4: Hyperparameter Sensitivity: The approach introduces a few new hyperparameters – notably the maximum number of slots N.**
>
> **The maximum number of slots N is indeed an important hyperparameter in object-centric models, but we would like to clarify that it is not introduced by our method**, as it is a standard assumption across all slot-based architectures.
> What our contribution adds is a semantic role for this parameter: by integrating objectness predictions directly into the symbolic reasoning and reconstruction processes, redundant or unused slots are naturally suppressed, reducing the sensitivity normally associated with choosing N.
> In practice, this mechanism makes the model more robust to imperfect choices of N, since non-object slots receive low objectness and have negligible impact on the final output. This behaviour is illustrated in Appendix C.4, where we visualize alpha masks together with their objectness scores.

---

### Official Review · Reviewer_r4Gj · 2025-10-31

**Soundness:** 3
**Presentation:** 3
**Contribution:** 2
**Rating:** 4
**Confidence:** 4

**Summary:**

The paper proposes DeepObjectLog, a neurosymbolic architecture that couples an object-centric perceptual module (Slot Attention) with probabilistic logic programming (ProbLog) to learn from distant supervision: only global task labels and a logic program, with no object-level annotations. The model is trained with an objective that maximizes a discriminative lower bound via amortized MAP inference. The method is evaluated on MultiMNIST-Addition, PokerRules, and CLEVR-Addition, targeting compositional, task, and object-count generalization. Reported results show notable gains over CNN/Slot-based baselines and a NeSy baseline, and better OOD performance than prompting VLMs (LLaVA-1.5, Qwen2.5-VL) on the MultiMNIST dataset.

**Strengths:**

1. The paper is well written, clear, and easy to follow.
2. The evaluations focus on out-of-distribution tasks, which are highly relevant and remain partly unresolved in recent years.

**Weaknesses:**

1. Fairly simple evaluation: (1) datasets are synthetic toy datasets, which is acknowledged in the limitations, and (2) the tasks are quite simple reasoning problems that can be expressed as a multiclass classification problem. I would like to see comparisons on (1) a more complex, general task such as VQA and (2) at least one real-world (or close to real-world) dataset.
2. Some claims in the paper seem not to be correct, and some of the main conclusions are already well known in the object-centric (OC) community. Please see the questions below for details.
3. The paper does not analyze or quantify robustness to imperfect/underspecified rules or label noise, nor disentangle how much gain comes from the rules vs. the OC module.

**Questions:**

1. In Line 72 it is stated that OC models often rely on object-level supervision, either a label or annotation for each object. This is not true. Slot Attention is a similarity-based method that works like a soft k-means [14], and is generally trained fully unsupervised with a decoder and a reconstruction loss. The only information Slot-Attention–based OC models require beforehand is the number of slots (i.e., objects), which has been a downside shown to cause over- or under-segmentation [1]. However, [2, 3] address this limitation and show on-par or better performance compared with plain Slot Attention. I would appreciate it if the authors could elaborate.
2. Line 131: it is claimed that without logical feedback, current OC models cannot learn from relational structure unless it is explicitly specified. That is not accurate. Several works evaluate OC models on a wide range of downstream reasoning tasks, from RL to VQA, in both in-distribution [4, 5] and out-of-distribution [6] settings, showing that OC representations capture relational information well and can be on par with foundation models for ID and compositional OOD performance.
3. I would like to see DINOSAUR-like OC models [7], where a pretrained backbone provides representations to Slot Attention and reconstruction is done in latent space. I would also like to see the performance of a plain foundation model (e.g., DINOv3) with a small projection layer for the task as another baseline. With these additions, I believe the paper would become more relevant to a larger community.
4. Several claims and benefits of the proposed approach are already well-known, and some are properties of Slot Attention which are well-known in the OC community. For example, test-time generalizability of Slot Attention to different numbers of objects and out-of-distribution generalization of OC models have been analyzed in [6, 13]. Furthermore, several works have shown the shortcomings of VLMs on compositional generalization tasks [8, 9, 10, 11, 12]. I would appreciate it if the authors could elaborate on these works and how the paper relates to these prior works. Additionally, please elaborate on what are the implications and novel insights of the work and the impact the authors would expect for the community.
5. Regarding weakness #3, how sensitive is performance to inaccurate or incomplete logic rules (e.g., missing predicates, conflicting constraints)? Also, related to weakness #1, how can such rules be defined in more general settings for tasks such as captioning or VQA, and is there a practical limit to how far we can go with neurosymbolic AI?

[1] Zimmermann, Roland S., et al. "Sensitivity of slot-based object-centric models to their number of slots." _arXiv preprint arXiv:2305.18890_ (2023).

[2] Fan, Ke, et al. "Adaptive slot attention: Object discovery with dynamic slot number." _Proceedings of the IEEE/CVF Conference on Computer Vision and Pattern Recognition_. 2024.

[3] Liu, Hongjia, et al. "MetaSlot: Break Through the Fixed Number of Slots in Object-Centric Learning." _arXiv preprint arXiv:2505.20772_ (2025).

[4] Mamaghan, Amir Mohammad Karimi, et al. "Exploring the Effectiveness of Object-Centric Representations in Visual Question Answering: Comparative Insights with Foundation Models." The Thirteenth International Conference on Learning Representations. 2025.

[5] Yoon, Jaesik, et al. "An investigation into pre-training object-centric representations for reinforcement learning." _arXiv preprint arXiv:2302.04419_ (2023).

[6] Kapl, Ferdinand, et al. "Object-centric representations generalize better compositionally with less compute." _ICLR 2025 Workshop on World Models: Understanding, Modelling and Scaling_. 2025.

[7] Seitzer, Maximilian, et al. "Bridging the gap to real-world object-centric learning." _arXiv preprint arXiv:2209.14860_ (2022).

[8] Huang, Irene, et al. "Conme: Rethinking evaluation of compositional reasoning for modern vlms." _Advances in Neural Information Processing Systems_ 37 (2024): 22927-22946.

[9] Hsieh, Cheng-Yu, et al. "Sugarcrepe: Fixing hackable benchmarks for vision-language compositionality." _Advances in neural information processing systems_ 36 (2023): 31096-31116.

[10] Wu, Xindi, et al. "Conceptmix: A compositional image generation benchmark with controllable difficulty." _Advances in Neural Information Processing Systems_ 37 (2024): 86004-86047.

[11] Li, Baiqi, et al. "Genai-bench: Evaluating and improving compositional text-to-visual generation." _arXiv preprint arXiv:2406.13743_ (2024).

[12] Kempf, Elias, et al. "When and How Does CLIP Enable Domain and Compositional Generalization?." _arXiv preprint arXiv:2502.09507_ (2025).

[13] Dittadi, Andrea, et al. "Generalization and robustness implications in object-centric learning." _arXiv preprint arXiv:2107.00637_ (2021).

[14] Locatello, Francesco, et al. "Object-centric learning with slot attention." _Advances in neural information processing systems_ 33 (2020): 11525-11538.

---

> ### Author Response · Authors · 2025-11-21
>
> We thank the reviewer for the constructive feedback and for highlighting the clarity of the presentation and the relevance of our focus on out-of-distribution evaluation. We appreciate the comments regarding the strengths and limitations of the work, and we address each of the raised questions below.
>
> &nbsp;
>
> **Question 1: In Line 72 it is stated that OC models often rely on object-level supervision, either a label or annotation for each object. This is not true. The only information Slot-Attention–based OC models require beforehand is the number of slots (i.e., objects), which has been a downside shown to cause over- or under-segmentation [1]. However, [2, 3] address this limitation and show on-par or better performance. I would appreciate it if the authors could elaborate.**
>
> We sincerely thank the reviewer for pointing this out. The current formulation in Line 72 is indeed imprecise and will be corrected. Our intention was to refer to object-centric models used for downstream classification or set-prediction tasks, where object-level supervision (e.g., labels or annotations for each object) is often required to map the discovered slots to semantic ground truth. For example, even the original Slot Attention paper includes a set-prediction task where ground-truth object labels are provided.
> However, this does not apply to, e.g., unsupervised object-centric models in a general way.
> **To avoid confusion, we will replace the original sentence with the following clearer and more accurate phrasing:**
>
> “While core object-centric models can be trained without annotations, pipelines that use these models for downstream semantic tasks often introduce object-level supervision to map each discovered slot to a symbolic category.”
>
> We also appreciate the reviewer’s remark regarding the slot-number limitation. Standard Slot Attention requires predefining the maximum number of slots, which may lead to under- or over-segmentation, and recent adaptive-slot methods [2,3] (reviewer’s cited works) offer compelling solutions. **In fact, our model builds directly on [2]** (as discussed in Section 3.3), but extends it by giving slots a task-dependent semantic meaning: each slot’s “objectness” is explicitly linked to the logical program. This provides a principled mechanism for suppressing redundant slots and propagating only meaningful object candidates to the symbolic layer. In this sense, **our approach inherits the benefits of adaptive-slot methods while integrating them into a reasoning-aware framework.**
>
> &nbsp;
>
> **Question 2: Line 131: it is claimed that without logical feedback, current OC models cannot learn from relational structure unless it is explicitly specified. That is not accurate. Several works evaluate OC models on a wide range of downstream reasoning tasks, from RL to VQA, in both in-distribution [4, 5] and out-of-distribution [6] settings, showing that OC representations capture relational information well and can be on par with foundation models for ID and compositional OOD performance.**
>
>
> We thank the reviewer for raising this point. The current phrasing in Line 131 does not fully capture the distinction we intended to make and will be revised. We fully agree to the comment. We take the opportunity to clarify our intended definition.
>
>
> The cited works indeed show that object-centric models can support downstream reasoning-like behaviour (e.g., through RL agents or foundation models) once an additional neural component is placed on top of the slot embeddings.
> However, our approach does explicitly incorporate logical (formal, interpretable, verifiable) reasoning into the learning of the perceptual model itself. The reasoning is not carried out by a separate neural module trained on object-centric features, but by an explicit symbolic program providing structured logical feedback. In other words, **in the cited works, the perceptual encoder never receives supervision that reflects logical constraints during training.**
> Our contribution concerns exactly this missing aspect: DeepObjectLog integrates a probabilistic logic program into the training loop, so that logical constraints guide the perceptual front-end end-to-end. This allows the model to learn object representations that are not only relationally structured, but aligned with the symbolic and human semantics of the task, a capability not addressed in prior OC pipelines.
>
> **To improve the clarity, we will update the sentence in the paper to:**
>
> “Without an additional task-level or logical supervision signal, current object-centric models learn object representations independently of the symbolic logic structure of the task, which limits their ability to identify objects, their relations, and their task-specific contributions in a way that aligns with human interpretation.”.

---

> ### Author Response · Authors · 2025-11-21
>
> **Question 3: I would like to see DINOSAUR-like OC models [7], where a pretrained backbone provides representations to Slot Attention and reconstruction is done in latent space. I would also like to see the performance of a plain foundation model (e.g., DINOv3) with a small projection layer for the task as another baseline. With these additions, I believe the paper would become more relevant to a larger community.**
>
> We thank the reviewer for this suggestion. We fully agree that evaluating pretrained  perception modules is important, and we conducted the requested experiments: a DINOSAUR-like OC model (ResNet-50 backbone + Slot Attention, latent-space reconstruction) and a DINOv2 encoder followed by a MLP classifier. Both backbones are pretrained on ImageNet. The results are summarized below:
>
> | Model |  Test | OOD | Extrap4 | Extrap5 |
> | -------- | ------- | -------- | ------- | ------- |
> | Dinov2 +MLP |  77.17 | 02.80 | 10.23 | 4.73 |
> | DINOSAUR-like with resnet50  |  97.50 | 10.50 | 27.33 | 7.67 |
> | Ours |  94.26 | 90.00  | 69.73 | 44.06 |
>
>
> These experiments were implemented carefully:
> for DINOv2 we followed the standard protocol (encoder + 2-layer MLP), exploring nine joint configurations of learning rates and hidden sizes;
> for the DINOSAUR-like baseline, we replicated the architecture as closely as possible within our setting, tuning both training parameters, hidden sizes and iterations of the slot attention module for a total of eighteen configurations.
>
> **Two clear conclusions emerge:**
>
> **Object-centric representations are indeed far more suited to these tasks** than global foundation-model features consistent with the reviewer’s cited works (i.e. [4,5] from previous question)
>
> **Logical feedback remains crucial:** even when paired with a stronger perceptual backbone or used in more complex architectures, purely neural OC models do not acquire the task-relevant relational structure needed for OOD generalization in our settings.
>
> &nbsp;
>
> **Question 4: Several claims and benefits of the proposed approach are already well-known, and some are properties of Slot Attention which are well-known in the OC community. For example, test-time generalizability of Slot Attention to different numbers of objects and out-of-distribution generalization of OC models have been analyzed in [6, 13]. Furthermore, several works have shown the shortcomings of VLMs on compositional generalization tasks [8, 9, 10, 11, 12]. I would appreciate it if the authors could elaborate on these works and how the paper relates to these prior works. Additionally, please elaborate on what are the implications and novel insights of the work and the impact the authors would expect for the community.**
>
> We thank the reviewer for the helpful pointers. The prior works cited indeed show that object-centric models can generalize in controlled settings. Our contribution is complementary: **when these models are trained directly for a downstream reasoning task using both reconstruction and task label, they do not exhibit the expected OOD generalization.** Our results show that, despite being given the same information, purely neural OC baselines fail to recover task-relevant relational structure, whereas adding symbolic constraints allows the perceptual module to align with the semantics required for reasoning. In other words, while the object-centric inductive bias enables OOD generalization in general, our model better exploits such capabilities.
>
> Regarding VLMs, our findings in Table 4 are consistent with the recent literature: large pretrained models struggle with compositionality. Although the comparison is not entirely fair, the intention of the experiment is to confirm that the benchmarks we consider are not trivial.
>
> **Implications.** DeepObjectLog does not claim new OC inductive biases; instead, it shows how symbolic reasoning can shape object-centric perception when trained end-to-end. This provides (i) a principled way to enforce task-relevant structure, (ii) a flexible interface for task transfer by simply changing the logic program, and (iii) an empirical refinement of expectations about what OC models can achieve without symbolic guidance.

---

> ### Author Response · Authors · 2025-11-21
>
> **Question 5: Regarding weakness #3, how sensitive is the performance to inaccurate or incomplete logic rules (e.g., missing predicates, conflicting constraints)? Also, related to weakness #1, how can such rules be defined in more general settings for tasks such as captioning or VQA, and is there a practical limit to how far we can go with neurosymbolic AI?**
>
> We appreciate the reviewer’s question. A central perspective guiding our work is that neurosymbolic systems are a strict superset of deep learning: when symbolic knowledge is available, it can be injected to guide learning and generalization, while in its absence the model naturally reduces to a purely neural architecture. In this sense, our approach does not place additional requirements on the user as it merely enables the use of knowledge when it exists. Such knowledge may come from experts, structured sources, or can even be extracted automatically with LLMs [1].Symbolic knowledge in our framework does not need to be fully consistent (because it is probabilistic) nor complete (because the neural component can compensate for missing information). **These are common critiques of symbolic systems, but here the symbolic program serves primarily as a descriptive specification of the task rather than a rigid prescriptive constraint.** It guides learning by articulating how the task should be interpreted, independently of the specific instances seen during training, and it provides a way to verify that the learned representations align with the intended human-level semantics. This alignment is precisely what enables effective OOD generalization, since the model can rely on the task’s structural description rather than memorizing patterns in the data.
>
> Our contribution focuses on the perceptual–symbolic interface: learning grounded object-centric representations that can support logical reasoning directly from raw input. How the symbolic rules themselves are obtained is orthogonal to this contribution. In practice, the rules we use are task-level specifications, not per-instance annotations, and the model does not constrain where they come from. Moreover, the symbolic layer is fully compatible with approaches that learn logic programs from data.
>
> Extending DeepObjectLog to handle partial or noisy rules is a promising avenue but beyond the scope of the present paper. For this, we can take inspiration from the field of concept-based learning [2], where recently neurosymbolic models have been developed that maintain their performance under imperfect symbolic abstractions and/or rules (e.g., [3,4,5]).
>
> **References:**
>
> [1] Shindo, Hikaru, et al. "Deisam: Segment anything with deictic prompting." Advances in Neural Information Processing Systems 37 (2024).
>
> [2] Koh, Pang Wei, et al. "Concept bottleneck models." International conference on machine learning. PMLR, (2020).
>
> [3] Debot, David, et al. "Interpretable concept-based memory reasoning." Advances in Neural Information Processing Systems 37 (2024)
>
> [4] Mahinpei et al. Promises and Pitfalls of Black-Box Concept Learning Models. 2021
>
> [5] Sawada, Yoshihide, and Keigo Nakamura. "Concept bottleneck model with additional unsupervised concepts." IEEE Access 10 (2022)

---

> > ### Comment · Reviewer_r4Gj · 2025-11-27
> >
> > Thanks for the thorough and thoughtful responses, clarifications, and the additional experiments. Most of my concerns have been addressed.
> >
> > A minor suggestion on the DINOSAUR-like baseline: I fully understand the time constraints during rebuttal, but for the camera-ready version, it would be interesting to repeat this experiment with a DINOvX (e.g. DINOv2 or DINOv3) backbone rather than ResNet-50, given that there is a considerable gap between the performance of the two.
> >
> > Regarding my earlier concern on real-world applicability (weakness #1 / Q5): I agree that this is largely an inherent limitation of current neurosymbolic approaches rather than of the present work. Overall, considering the rebuttal and the other reviewers’ comments, I now lean towards the acceptance of the paper.

---

### Official Review · Reviewer_mfP4 · 2025-11-01

**Soundness:** 3
**Presentation:** 3
**Contribution:** 3
**Rating:** 6
**Confidence:** 3

**Summary:**

This paper aims to integrate a neurosymbolic reasoning component into a object-centric architecture and learn the model in an end to end fashion. To this end, the authors leverage an object centric approach which yields explicit uncertainty estimates over the presence and class of an object and integrate it with the ProbLog probabilistic programming framework. On several logical reasoning task across a few simple image datasets, the authors show superior logical reasoning performance using this model relative to baseline approaches.

**Strengths:**

* The paper is very well written and is easy to follow.

* The paper tackles an ambitious problem, namely, performing logical reasoning over symbols by leveraging object-centric representations.

* Integrating ProbLog with a slot-based neural network model is novel to the best of my understanding and is an interesting approach.

* The experimental section is well presented and the results showing the benefits of DeepObjectLog are convincing.

**Weaknesses:**

* I found the presentation in Section 3.4 a bit confusing regarding how ProbLog is integrated into the model. Is it the case that the logical rule/task to be performed, e.g., add the two numbers in the image, is defined a priori, or is the task also inferred from observed data.

* I can imagine DeepObjectLog working well in situations in simple logical reasoning task as the authors tested, however, I am skeptical of how the method will perform for more complex reasoning task in which the symbols and algorithm/logical program to be executed are less straightforward to define. For example, how do the authors imagine their method would perform on the ARC challenge [1]?

* The authors rely on a masked decoder which poses a limitation in terms of scalability. I believe it is important to understand whether DeepObjectLog can leverage more complex decoders such as the Transformer decoders in [2, 3].

* I believe the statement "However, these models often rely on object-level supervision, meaning they require a label or annotation for each object in the image" in line 72 is incorrect. All of the models the authors cite in the previous sentence are unsupervised.

**Questions:**

* Is the logical rule/program hard coded for a given task or is it inferred from the task?

* How do the authors imagine their approach would work for more complex reasoning task such as the ARC challenge?

* Do the authors believe that more scalable decoders can be integrated into their approach?

**References**

[1] Chollet et. al 2019, On the Measure of Intelligence

[2] Singh et. al 2021, Illiterate DALL-E Learns to Compose

[3] Brady et. al, 2024, Interaction Asymmetry: A General Principle for Learning Composable Abstractions

---

> ### Author Response · Authors · 2025-11-21
>
> We sincerely thank the reviewer for their thoughtful and positive assessment of our work, especially for highlighting the clarity of the paper, the ambition of the problem we tackle, and the novelty of our integration. We address below the reviewer's questions and concerns.
>
> &nbsp;
>
> **Weakness 4: I believe the statement in line 72 is incorrect. (“However, these models
> often rely on object-level supervision, meaning they require a label or annotation for each object in the image”)**
>
> We sincerely thank the reviewer for pointing this out. The current formulation in Line 72 is indeed imprecise and will be corrected. Our intention was to refer to object-centric models used for downstream classification or set-prediction tasks, where object-level supervision (e.g., labels or annotations for each object) is often required to map the discovered slots to semantic ground truth. For example, even the original Slot Attention paper includes a set-prediction task where ground-truth object labels are provided.
> However, this does not apply to, e.g., unsupervised object-centric models in a general way.
> **To avoid confusion, we will replace the original sentence with the following clearer and more accurate phrasing:**
>
> “While core object-centric models can be trained without annotations, pipelines that use these models for downstream semantic tasks often introduce object-level supervision to map each discovered slot to a symbolic category.”
>
> &nbsp;
>
> **Question 1: Is the logical program hard-coded or inferred?**
>
> As described in our problem formulation, DeepObjectLog requires specifying task-level logical rules once per task. These rules describe how object-level concepts should be composed to determine the global label, but it does not provide any per-instance supervision. **This is standard in neurosymbolic learning and reflects the premise of the paradigm: when symbolic knowledge is available, it should be seamlessly usable by the model.**  If no knowledge is provided, the model trivially reduces to a purely neural architecture, as standard deep approaches are simply a special case within our framework. Conversely, when partial knowledge exists (e.g. from human experts, domain rules, knowledge bases, or even rules induced by LLMs [1]), the same interface allows it to be incorporated naturally and consistently. The symbolic program is therefore reusable and does not need to be rewritten when the number of objects changes or when scenes contain more (or fewer) elements than those seen during training.
> **At the same time, our model does not preclude learning the rules:** our base paradigm  (DeepProbLog) supports rule learning via parameter learning, and rule-induction methods (e.g. ILP systems [2]) can in principle generate the symbolic component automatically. Our contribution is complementary: we focus on learning grounded, object-centric representations that can support reasoning regardless of whether the symbolic program is hand-written or learned.
>
> &nbsp;
>
> **References:**
>
> [1] Shindo, Hikaru, et al. "Deisam: Segment anything with deictic prompting." Advances in Neural Information Processing Systems 37 (2024).
>
> [2] Cropper, Andrew, et al. "Inductive logic programming at 30." Machine Learning 111.1 (2022)

---

> ### Author Response · Authors · 2025-11-21
>
> **Question 2: How do the authors imagine their approach would work for more complex reasoning task such as the ARC challenge?**
>
> We see challenges like ARC as natural and ambitious directions that highlight why the problems we study matter. Such benchmarks rely on identifying and manipulating objects in unstructured grids, performing simple algorithmic transformations (e.g., counting, sorting, comparing quantities), and applying these transformations out of distribution. These aspects resonate with the abilities our method aims to develop: learning object-centric representations from raw input and executing symbolic programs beyond the specific configurations seen during training.
>
> At the same time, challenges of this kind also require something our work does not attempt: inferring both the symbolic vocabulary and the rule itself from only a few examples. While there is substantial literature on inducing rules or program structures from examples, comparatively little attention has been given to learning meaningful representations in such weakly structured scenarios. DeepObjectLog advances this latter direction. **It does not solve the full problem posed by challenges like ARC, but we believe it represents a meaningful step toward the perceptual–symbolic integration required to eventually tackle them.**
>
> &nbsp;
>
> **Question 3: Do the authors believe that more scalable decoders can be integrated into their approach?**
>
> **Yes, our approach is fully compatible with more scalable or expressive decoders.** The masked decoder we employ follows the standard Slot Attention formulation and was chosen for consistency with the baselines and prior work, not because the method fundamentally requires this specific architecture. In DeepObjectLog, the decoder operates independently from the symbolic layer, and its only interface with the reasoning is via the objectness scores and slot embeddings. This modularity means that any decoder capable of reconstructing scenes from slot-wise representations could be integrated. In principle, this includes Transformer decoders, diffusion-based decoders, or other generative architectures that scale more favorably with image complexity. The symbolic reasoning component would remain unchanged, as it depends only on the slot embeddings and objectness variables, not on the decoder structure.
>
> **To provide a concrete verification we did a new experiment**, replacing the masked CNN decoder with **a Transformer-based decoder** and evaluated the full model. We designed the integration so that the decoder operates directly on the slot representations produced by Slot Attention, projecting each slot into a token sequence and decoding it into an image contribution and ensuring full compatibility with the existing object-centric interface.
> Even with this non-specialized decoder, the model still reached 90% OOD accuracy on MNIST-Addition, well above all neural baselines.

---

> > ### Comment · Reviewer_mfP4 · 2025-11-27
> >
> > Dear authors,
> >
> > Thank you for your detailed feedback to my comments and for the additional experiments.
> >
> > The main concern I have with the approach is that, as the authors clarified, the symbolic rule must be known a priori. I think this poses a core limitation in the scalability of this approach, and I encourage the authors to include a deeper discussion of this limitation. Regardless, however, I agree with the authors that inferring the symbolic rule/algorithm is a more ambitious and challenging problem and need not be solved in this paper, for the work to have value.
> >
> > To this end, I am happy to recommend acceptance and have changed my score to reflect this more directly.

---

### Official Review · Reviewer_1DeD · 2025-11-02

**Soundness:** 3
**Presentation:** 3
**Contribution:** 2
**Rating:** 6
**Confidence:** 5

**Summary:**

This work proposes an object-centric neurosymbolic approach that can be trained end-to-end (i.e. via distant supervision). By combining and object-centric visual encoder with a probabilistic symbolic reasoning framework, the model combines the perceptual strength of neural networks with the reasoning strength of symbolic models, without the need for intermediate supervision of perceptual representations (which is required by most neurosymbolic approaches). The approach is tested on an MNIST arithmetic task and classification of poker hands, demonstrating improved out-of-distribution generalization.

**Strengths:**

- The proposed model overcomes one of the primary weaknesses of neurosymbolic approaches, which is that they typically depend on very detailed object-level annotations to train the perceptual encoder. This approach more deeply integrates the neural/perceptual and symbolic/reasoning components, enabling it to be trained based on downstream task error.
- The model demonstrates promising results on various out-of-distribution generalization settings.
- The paper is very clearly written, and nicely frames the previous work and challenges in this area.

**Weaknesses:**

- The primary limitation is that only synthetic tasks are investigated. Additionally, these tasks both involve relatively simple classification, so they do not provide the strongest test of the symbolic component's reasoning abilities. The CLEVR-addition dataset is a step in the right direction, but this task still involves simple, synthetic images and limited (one-step) reasoning. It would be more compelling if the model could be extended to tasks that require multi-step reasoning and inference, and more complex, naturalistic images.
- Regarding the attention masks shown in the supplementary material, first it might be better to visualize these by applying the attention weights to the input image, or by showing the slot-specific reconstructions. Second, the results suggest that the model may not be as object-centric as is desired. In particular, for conditions involving one or just a few objects, it looks like multiple slots are being used to encode the same object.

**Questions:**

- Can the proposed approach be readily extended to more complex settings?
- To what extent are the learned representations actually object-centric?

---

> ### Author Response · Authors · 2025-11-21
>
> We sincerely thank the reviewer for highlighting the significance of our contribution toward addressing a key limitation in neurosymbolic approaches, as well as for recognizing the strength of our perceptual–symbolic integration and the clarity of the work. We address the raised points below.
>
> &nbsp;
>
> **Question 1: Can the approach be extended to tasks that involve more complex reasoning (i.e. beyond classification)?**
>
> **The proposed framework is in fact compatible with more complex reasoning settings, and we see this as a highly promising direction.**  One of the strengths of our framework is its use of a full probabilistic logic programming language (ProbLog) as the symbolic layer. This gives our system the expressive power to model not only classification tasks but also complex, multi-step reasoning processes such as chaining and deduction. Crucially, in our setting, increasing the reasoning complexity typically means modifying the logic rules rather than changing the architecture. The object-centric representations and their objectness scores are extracted as usual, and the symbolic layer can reason over these with more expressive rules. As such, the learning dynamics remain unchanged: supervision still comes from downstream task-level labels, and the model can propagate this signal back through the entire reasoning process and down the perception layer.
>
> That said, extending our system to more complex scenarios may introduce new challenges. For example, as task supervision becomes even more sparse or indirect, the risk of learning reasoning shortcuts [1] could increase. Another challenge lies in the capabilities of the perceptual models: most object-centric architectures (e.g., Slot Attention) still struggle with cluttered or real-world scenes. However, our model is modular by design and can incorporate any object-centric extractor. This means improvements in these perceptual components will directly benefit our framework without requiring architectural changes.
>
> &nbsp;
>
> **Weakness: Regarding the attention masks shown in the supplementary material, first it might be better to visualize these by applying the attention weights to the input image, or by showing the slot-specific reconstructions.**
>
> We thank the reviewer for the suggestion. **We've included (in the appendix of the revised document) a new version of the alpha mask visualization**, where the masks are applied to the input images, making it clearer how they interact with them in them and the objectness score.
>
> &nbsp;
>
> **References:**
>
> [1] Marconato, Emanuele, et al. "Not all neuro-symbolic concepts are created equal: Analysis and mitigation of reasoning shortcuts." Advances in Neural Information Processing Systems 36 (2023)

---

> ### Author Response · Authors · 2025-11-21
>
> **Question 2: To what extent are the learned representations actually object-centric? In particular in conditions involving one or just a few objects, it looks like multiple slots are being used to encode the same object.**
>
> We appreciate this important question and take the opportunity to clarify our definition of object-centricity and how it is enforced in our model.
> In standard object-centric architectures, objectness typically emerges from architectural biases: slots are encouraged to specialize through iterative attention updates, and mutual exclusivity is induced via the Softmax over attention maps. Slot Attention is a well-known example: it learns to separate entities without supervision, but nothing prevents two slots from partially encoding the same object unless this is discouraged implicitly through reconstruction.
> **We model an explicit objectness variable which is used by the logic and the decoder.**
> In our model, object-centricity is shaped not only by architectural priors but also by an explicit objectness variable that is tightly integrated in both the probabilistic logic program and the decoder.
> As explained in the “ProbLog for object-centric learning” paragraph, each slot receives a learned probability of being a valid object. This probability influences:
>  (i) symbolic reasoning (only slots with high objectness meaningfully contribute to logical outcomes), and
>  (ii) visual reconstruction (objectness gates the alpha-mask contribution of each slot).
>
> **The object detection process is then not only guided by visual features but also by logical coherence.** For example, a sum greater than 9 cannot be reached by a single object in MNIST: therefore, detection of a single object in such a scene is discouraged by the logic (through backpropagation through the program).
> Crucially, this mechanism also improves behavior in scenes with fewer objects than slots. Redundant or overlapping slots naturally receive low objectness scores and therefore have negligible impact on both the reasoning module and the reconstruction. This prevents multiple slots from encoding the same region and provides an additional guiding signal that helps resolve ambiguities that are otherwise difficult for purely neural object-centric models to handle.
>
> As shown in Appendix C.4, **the visualized objectness scores make this effect explicit:** slots corresponding to meaningful entities receive high objectness (“Yes” in the figure) and contribute to the logic program and the masked decoder, while redundant slots are systematically suppressed.

---

> > ### Comment · Reviewer_1DeD · 2025-11-26
> >
> > Thank you to the authors for these replies. I appreciate the updated figures visualizing the attention masks and the role of the objectness score. I think that the synthetic nature of the tasks investigated is still a limitation. I appreciate that the framework can in principle handle more complex tasks, but showing that this actually works in practice is important for establishing the viability of the approach, especially given some of the concerns that the authors highlight (e.g. reasoning shortcuts). On balance I still support acceptance, but I think the work would be stronger with the incorporation of more complex reasoning tasks.

---

### Author Response · Authors · 2025-12-02

Dear AC,

From the authors’ perspective, the discussion phase led to the resolution of the main technical concerns for the reviewers, with consistent feedback that the clarifications and additional experiments strengthened the paper. We want to highlight the final conclusions of the reviewers after rebuttal:

| **Reviewer** |  **Initial score** | **Comment after the rebuttal** |
| -------- | ------- | -------- |
| **1DeD** |  **6** | “On balance **I still support acceptance**, but I think the work would be stronger with the incorporation of more complex reasoning tasks.“ |
| **mfP4**   |   **6** | “To this end, **I am happy to recommend acceptance and have changed my score to reflect this more directly.**” |
| **r4Gj**  |   **4** | “Overall, considering the rebuttal and the other reviewers’ comments, **I now lean towards the acceptance of the paper.**”  |
| **Nexu**  |   **6** | Discussion was concluded before they could respond.  |

&nbsp;

We also provide the following summary to clarify how the main concerns raised during the review process were addressed during the discussion phase.

&nbsp;

**Reviewer 1DeD** focused primarily on the complexity of the evaluated reasoning tasks and on whether the learned representations are genuinely object-centric. In response, we clarified that the proposed framework naturally supports more complex, multi-step logical reasoning without architectural changes, while also acknowledging the current limits imposed by object-centric perception in cluttered or real-world scenes. To directly address the object-centricity concern, we provided a detailed explanation of the role of explicit objectness variables and added new visualizations in the appendix showing how redundant slots are suppressed. **In their follow-up, the reviewer acknowledged the improved explanations and figures.**

&nbsp;

**Reviewer mfP4** raised concerns about whether symbolic rules must be predefined, about scalability of the decoder, and about extensibility to challenges such as ARC. We clarified that task-level symbolic programs are specified once per task following standard NeSy practice as stated in our problem definition, but that our framework remains fully compatible with rule learning and ILP approaches. We also emphasized that our contribution focuses on grounding symbolic interfaces in perception. To address scalability of the decoder, we conducted an additional experiment with a Transformer-based decoder and showed that the model still achieves strong OOD performance. **In their final response, the reviewer stated that their primary concerns had been resolved.**

&nbsp;

**Reviewer r4Gj** questioned several claims regarding supervision in object-centric learning, the necessity of logical feedback, and requested stronger baselines using pretrained backbones. We corrected the imprecise statement on object-level supervision and revised it in the manuscript. We also clarified the distinction between downstream neural reasoning on frozen object-centric features and our end-to-end logic-guided training. To address the baseline request, we added **new experiments** with a DINOSAUR-like model and a DINO-based classifier, carefully tuned across multiple configurations. These new results further supported our central claim that symbolic feedback during training is essential for robust OOD generalization. **In their final response, the reviewer indicated that most concerns were addressed and that the remaining ones are mostly related to the current neurosymbolic AI than to our method.**

&nbsp;

**Reviewer Nexu** focused on the introduction of novel logical classes at test time, computational scalability, source of performance gains, and sensitivity to the number of slots. We clarified that new logical rules are introduced purely at test time without retraining, enabled by the clean separation between perception and reasoning. On scalability, we explained our use of state-of-the-art inference backends and demonstrated reasoning over up to 14 objects per sample in CLEVR-Addition, with an additional plot added in the appendix. On ablations, we clarified that Slot Attention and CoSA already serve as neural-only counterparts and that a detailed component-wise ablation is included in the appendix. For the slot hyperparameter, we explained how objectness reduces sensitivity to N. **This reviewer did not provide a post-rebuttal response but raised no further objections after the clarifications.**

&nbsp;

We thank the reviewers for their time and constructive feedback, which contributed to improving the paper.

&nbsp;

Best Regards,

The Authors

---

### Meta-Review · Area_Chair_Xgym · 2026-01-06

**Summary:**

First of all, the clarity.

And:

Only synthetic tasks are investigated. These tasks both involve relatively simple classification

Some claims in the paper seem not to be correct, and some of the main conclusions are already well known in the object-centric (OC) community, e.g., incorrect statement: "However, these models often rely on object-level supervision, meaning they require a label or annotation for each object in the image" in line 72.

Scalability and Complexity: the symbolic rule must be known a priori, which is impractical.

Does not analyze or quantify robustness to imperfect/underspecified rules or label noise, nor disentangle how much gain comes from the rules vs. the OC module.

**Reviewer Concerns:**

Through the discussions, the scalability and application of the proposed method, and the toy syn experiment problem were not addressed.

**Reviewer Scores:**

After reading the paper and discussion, I agree that the proposed concerns, especially the experiment design and claim problem. One of the negative reviewers may have raised the score. However, the whole score picture still leans towards a rejection, in my opinion. I believe this paper still needs to add more solid tasks and results to support its claims before acceptance.

---

### Decision · Program_Chairs · 2026-01-26

Reject